# Impact of seawater sulfate concentration on sulfur concentration and isotopic composition in calcite of two cultured benthic foraminifera

Caroline Thaler[1,2], Guillaume Paris[3], Marc Dellinger[2], Delphine Dissard[4], Sophie Berland[5], Arul Marie[2], Amandine Labat[2], Annachiara Bartolini[1]

[1]CR2P UMR 7207 MNHN CNRS SU, F-75005 Paris France
[2]MCAM UMR 7245 MNHN CNRS, F-75005 Paris France
[3]Université de Lorraine-CNRS, CRPG UMR 7358, F-54000 Nancy France
[4]LOCEAN UMR 7159 IRD SU CNRS MNHN, F-75005 Paris France/ Nouméa New Caledonia
[5]BOREA UMR 8067 MNHN CNRS SU, F-75005 Paris France

*Correspondence to*: Caroline Thaler (thaler.caroline@gmail.com)

**Abstract.** Marine sediments can be used to reconstruct the evolution of seawater $[SO_4^{2-}]$ and $\delta^{34}S$ over time, two key parameters that contribute to refine our understanding of the sulfur cycle and thus of Earth's redox state. $\delta^{34}S$ evolution can be measured from carbonates, barites and sulfate evaporites. $[SO_4^{2-}]$ variations can be reconstructed using fluid inclusions in halites, a method that only allows a low-resolution record. Reconstruction of the past sulfur cycle could be improved if carbonates allowed to track both seawater $\delta^{34}S$ and $[SO_4^{2-}]$ variations in a sole, continuous sedimentary repository. However, most primary carbonates formed in the ocean are biogenic, and organisms tend to overprint the geochemical signatures of their carbonates through a combination of processes often collectively referred to as vital effects. Hence, calibrations are needed to allow seawater $\delta^{34}S$ and $[SO_4^{2-}]$ reconstructions based on biogenic carbonates. Because foraminifera are important marine calcifiers, we opted to focus on calcite synthesized by individuals of rosalinid benthic foraminifera cultured in laboratory under controlled conditions, with varying seawater $[SO_4^{2-}]$ (ranging from 0 mM to 180 mM). Our experimental design allowed us to obtain foraminiferal asexual reproduction over several generations. We measured bulk Carbonate Associated Sulfate (CAS) content and sulfur isotopic composition ($\delta^{34}S_{CAS}$) on samples of tens to hundreds of specimens from a selection of culture media, where $[SO_4^{2-}]$ varied from 5 to 60 mM. Increasing or decreasing $[SO_4^{2-}]$ with respect to modern-day seawater concentration (28 mM) impacted foraminiferal population size dynamics and the total amount of bioprecipitated carbonate. Foraminiferal CAS concentration increased proportionally with $[SO_4^{2-}]$ concentration from 5 mM up to 28 mM, and then showed a plateau from 28 to 60 mM. The existence of a threshold at 28 mM is interpreted as the result of a control on the precipitation fluid chemistry that foraminifera exert on the carbonate precipitation loci. However, at high seawater sulfate concentrations (> 40 mM) the formation of sulfate complexes with other cations, may partially contribute to the non-linearity of the CAS concentration in foraminiferal tests at high increases in $[SO_4^{2-}]$. Yet, despite the significant effect of seawater $[SO_4^{2-}]$ on foraminiferal reproduction and on CAS incorporation, the isotopic fractionation between CAS and seawater remains stable through varying seawater $[SO_4^{2-}]$. Altogether, these results illustrate that CAS in biogenic calcite could constitute a good proxy for both seawater $[SO_4^{2-}]$ and $\delta^{34}S$ and suggests that sulfate likely plays a role in foraminiferal biomineralization and biological activity.

## 1 Introduction

In the modern ocean, marine organisms control the precipitation of most calcium carbonates through the biomineralization of calcite or aragonite, the two main $CaCO_3$ polymorphs. Biogenic calcium carbonates from the sedimentary record have been used for decades to reconstruct past environmental conditions. At modern sulfate and magnesium concentrations in seawater (about 28

mM and 50 mM respectively), aragonite precipitates preferentially over calcite in abiotic conditions at room temperature (Bots et al., 2011; Barkan et al., 2020; Goetschl et al., 2019). Seawater sulfate and magnesium concentrations varied over the last 550 Myr, ranging from ~5 mM to ~28 mM (Horita et al., 2002), and from ~44 to ~55 mM, respectively (Lowenstein et al., 2001; Brennan et al., 2004). Lower and higher seawater sulfate and magnesium concentrations have been shown to match calcitic and aragonitic oceans, where calcite or aragonite forming organisms were favored respectively (Lowenstein et al., 2003; Algeo et al., 2015; Lin et al., 2018, Goestchl et al., 2019). In the modern aragonitic ocean (Sandberg et al., 1983) as well as through parts of the geological past of Earth's history, the occurrence of calcitic organisms (e.g. foraminifera, coccolithophorids, some mollusks, bryozoans and coralline algae) could thus appear as a paradox. These calcitic organisms growing in aragonite oceans with high sulfate content would then have developed adaptive strategies and exerted a high degree of biological control in calcite bioprecipitation and sulfate incorporation, which need to be better understood.

Among the main calcite synthesizers, foraminifera are unicellular eukaryotes that build mainly calcitic (rare aragonitic species exist) shells named "tests", that accumulate on the ocean seafloor (Schiebel 2002; Langer 2008). As foraminifera build their tests, elements present in seawater get incorporated as traces in the biomineral structure. Sulfur is assumed to be incorporated in the calcium carbonate lattice structure as $SO_4^{2-}$ by replacing a $CO_3^{2-}$ group (Kontrec et al. 2004; Fernandez-Diaz et al. 2010) and is referred to as CAS, for Carbonate Associated Sulfate. This has been illustrated by an increase in S/Ca in benthic foraminiferal calcite as a function of seawater $[CO_3^{2-}]$ decrease (van Dijk et al., 2017). Paris et al. (2014) evidenced that the planktic species *Orbulina universa* faithfully records the $[SO_4^{2-}]$ /$[Ca^{2+}]$ ratio of the seawater in which it grew for $[SO_4^{2-}]$ values from 18 mM to 28 mM. These encouraging results, however, needed to be tested on benthic species and on a wider range of $[SO_4^{2-}]$, to cover deep time oceanic values, which varied from less than 5 mM up to 28 mM nowadays (Algeo et al. 2015) and potentially beyond, during important volcanic events in the past, or in the vicinity of sulfate-rich volcanic hydrothermal fluids on the seabed (Gamo et al. 1997; Laakso et al. 2020). Furthermore, the possibility that foraminiferal calcite could serve both as $[SO_4^{2-}]$ and $\delta^{34}S$ record needs to be further validated. While so far measurements in biogenic carbonates showed that sulfur isotopes are systematically fractionated by $\pm$ 1 ‰ from seawater (Kampschulte et al., 2001; Paris et al., 2014; Present et al., 2015; Rennie et al., 2018), recent experiments of abiotic $CaCO_3$ precipitation showed that a 2-5 ‰ fractionation of sulfur isotopes between aqueous sulfate and CAS in calcite covary with $[SO_4^{2-}]$ and, to a lesser extent, with precipitation rates (Barkan et al., 2020). There is thus a contrasting abiotic-biotic behavior that needs to be elucidated in order to determine whether calcitic foraminiferal tests could be used as a paleoenvironmental archive for the sulfur cycle, and suggests the possibility that seawater $[SO_4^{2-}]$ variations impact foraminiferal biocalcification and carbonate production.

To answer these questions, we grew two strains of Rosalinidae (Fig. 1), which are asymbiotic benthic foraminifera, at constant temperature, pH and salinity over a range from 0 to 180 mM of seawater $[SO_4^{2-}]$. Compared to planktic foraminifera, benthic foraminifera have two advantages: (i) they cover deeper geological times and (ii) they can reproduce more easily in experimental conditions.

In general, in both planktic and benthic foraminiferal culture experiments performed to calibrate geochemical proxies, populations of individuals captured in the wild do not have the time to adapt to the experimental conditions because maintaining foraminiferal reproductions over several generations is a complicated task. Therefore, measurements of geochemical proxies are usually performed either on the few test chambers that precipitated in the experimental medium (e.g. Dissard et al., 2010 a and b; van Dijk et al., 2017; Schmidt et al., 2022), or on whole tests that include the initial chambers grown in the natural marine environment prior to collection (e.g. Paris et al., 2014; Le Houedec et al., 2021) and account for this in some way, e.g. using size-mass relationships or labels. Our experiment was carefully designed to obtain several generations grown over several weeks in each experimental medium, ensuring both acclimatization and full precipitation of the test in the medium. Only live individuals that had fully grown

under the experimental conditions were collected for analysis, as empty shells of dead individuals were discarded at each previous
water change. We analyzed population size dynamics, as well as shell $\delta^{34}S$ and $[SO_4^{2-}]$ in bulk samples of tens to hundreds of
specimens in each medium to shed light on the mechanisms of sulfate incorporation in benthic foraminiferal calcite.

## 2 Materials and methods

### 2.1 Culture conditions

#### 2.1.1 Long term culture with asexual reproduction

Culture experiments were conducted at the French National Museum of Natural History (Muséum national d'Histoire naturelle,
MNHN) in the free living protist collection facilities (collection group: Biological Resources of Living and Cryopreserved Cells;
Collection of Unicellular Eukaryotes) on two previously cloned foraminiferal strains adapted to *in vitro* cell culture in 90 mm
diameter Petri dishes with natural sea water (NSW) and fed with *Chlorogonium sp.* (specimen MNHN-CEU-2016-0001), a
freshwater microalga. Two strains namely, For1C1 (specimen MNHN-CEU-2016-0075) and C1Tg (specimen MNHN-CEU-2016-
0075) (Fig. 1 and Fig. 2), were isolated from the top layer of sediments collected from Banyuls sea shore (Mediterranean French
coast) in 2006, and Concarneau (Atlantic French coast) in 2011, respectively.

Both strains were maintained through asexual reproduction (Fig. 3), using the following method: foraminifera were cultured in 90
mm diameter Petri dishes filled with 0.22 µm filtered NSW from Banyuls, France for For1C1 strain, or Concarneau, France for
C1Tg strains (Fig. 1).

The NSW was kept in a cold room for at least a month and its pH (NBS scale) was adjusted to 8.2 through addition of NaOH
and/or HCl, before use. The Petri dishes were kept at 22°C in an incubator (Memmert IPP 110 plus) equipped with cold white light
modules (5,500 K) with a 12h day-12h night cycle. Water in the Petri dishes was changed once a week and foraminifera fed with
living freshwater algae (*Chlorogonium* sp.). The algae were cultured in Basal Bold medium at 25°C under medium light intensity,
and suspended in sterile pH 8.2 NSW after 3 steps of rinsing with NSW. Live algae can have a major impact on the seawater
carbonate chemistry system by reproducing and consuming $CO_2$ through photosynthesis. As freshwater algae, the *Chlorogonium*
cells died immediately in seawater, without undergoing lysis. This prevents those not eaten by foraminifera from spreading and/or
being metabolically active and thus they do not influence the seawater chemistry conditions within the Petri dishes. The use of live
freshwater instead of seawater algae to feed foraminifera is therefore an innovative approach that is particularly suited to long term
culture experiments for the calibration of foraminiferal geochemical proxies, where seawater chemical conditions must be kept
under control. Every other week a new Petri dish was set up with a dozen of new juveniles (pre-adults below the age of asexual
reproduction, characterized by test with ~ 10 chambers). Live cultures were discarded after a month to prevent bacterial or fungal
spread.

#### 2.1.2 Culture in artificial seawater with varying $[SO_4^{2-}]$

In 2016, the two foraminiferal strains (For1C1 and C1Tg) were transferred to 0.22 µm filtered artificial seawater (ASW) mimicking
NSW (Fig. 1). The ASW was prepared following Kester et al. 1967. The total salinity was 35.06 g/L and the main ionic
concentrations were as follow, in mM: $Cl^-$ 543.9, $Na^+$ 467.3, $SO_4^{2-}$ 28.2, $Mg^{2+}$ 53.1, $Ca^{2+}$ 9.9, $K^+$ 10.0, $HCO_3^-$ 2.3, $Br^-$ 0.8, $H_3BO_3$
0.4, $Sr^{2+}$ 0.1, $F^-$ 0.1. After equilibration with the atmosphere for 2 to 3 hours, the pH of ASW was adjusted to pH 8.2 by the addition

of NaOH and HCl. ASW and NSW were sterilized by filtration on a 0.22 µm filter. The acclimation to ASW lasted approximately for a year (with foraminifera being transferred to new 90 mm diameter Petri dishes monthly) without any noticeable effect on the foraminiferal life cycle and morphology. Over this period of time, batches of several hundreds of foraminifera of each species (strains For1C1 and C1Tg), cultured either in ASW or NSW, were sampled for $[SO_4^{2-}]$ and $\delta^{34}S$ composition measurements. The C1Tg strain was only used for $[SO_4^{2-}]$ and $\delta^{34}S$ composition measurements of specimens from media in ASW and NSW at the current seawater average $[SO_4^{2-}]$ of 28 mM, whereas the For1C1 strain was also used for $[SO_4^{2-}]$ and $\delta^{34}S$ composition measurements of specimens from media with different $[SO_4^{2-}]$.

To produce media with different $[SO_4^{2-}]$, we created an ASW without $SO_4^{2-}$ (hereafter ASW[0]) and another with $[SO_4^{2-}]$ = 180 mM (hereafter ASW[180]). The amount of NaCl in those two media was adjusted to keep the total salinity constant (35.06 g/L). $Na^+$ concentrations for ASW[0] and ASW[180] were 479 mM and 402 mM, respectively, representing a maximum 24% change, while the $Cl^-$ concentrations were 612 mM and 175 mM, respectively, representing a maximum 71 % change, for a maximum 180 % change in sulfate concentration.

ASW[0] and ASW[180] were mixed in various proportions in order to obtain 8 other ASW with the following $[SO_4^{2-}]$: 1, 5, 10, 35, 40, 50, 60, 90 and 120 mM. Each of these media had the same salinity as ASW (35.06 g/L), pH (8.2), DIC (Dissolved Inorganic Carbon: $[CO_2]$+ $[H_2CO_3]$+$[HCO_3^-]$+$[CO_3^{2-}]$) and ALK (Alkalinity). For1C1 was the sole strain grown under different $SO_4^{2-}$ concentrations (Fig. 1).

In a first set of experiments (Set 1), 22 to 31 For1C1 individuals (Fig. 1, Table 1) with ~ 10 chambers each were transferred from ASW to new 60 mm diameter Petri dishes filled with the following media: ASW (hereafter ASW[28]), ASW[0], ASW[5], ASW[10], ASW[60], ASW[120] and ASW[180] and then cultured for 34 days. In parallel, 17 individuals so far cultured in natural seawater were moved to a new 60 mm diameter Petri dish containing NSW from Banyuls and were cultured for 39 days.

In a second set of experiments designed to refine the concentration step between 0 and 90 mM (Set 2), 6 individuals of For1C1, also presenting ~10 chambers each, were transferred from ASW to new 60 mm diameter Petri dishes and were cultured for 33 days in the following media ASW[28], ASW[1], ASW[10], ASW[35], ASW[40], ASW[50], and ASW[90] (Fig. 1, Table 1).

For populations of more than approximately 300 individuals, as obtained in media with concentration ranging from 5 to 35mM sulfate, the specimens were distributed over several 60 mm diameter Petri dishes (up to 3) to avoid problems associated with overpopulation. In both sets of experiment, *Chlorogonium* fed to foraminifera were rinsed and suspended in the media corresponding to each Petri dishes prior to their addition. Each week, live individuals were counted in each environment. As the species under study remains attached to the substrate when alive, individuals that no longer adhered to the Petri dishes, and therefore floated, were considered dead and not counted. However, it should be noted that rare dead individuals, which can sometimes be identified as unequivocally empty tests or as individuals without reticulopodial activity and/or change in color for several days (Bernhard, 2000), may remain attached and might have been counted as alive. At the same time, a few living adults may also have become detached from the substrate, and were therefore not counted as alive. As these cases are rare, the error generated by these two phenomena is largely covered by the error bar on the count in growing samples, while in samples not growing from the inoculum, living cells remain estimated. After counting, we sampled 6 mL of water through a 0.2 µm filtered for DIC and $[SO_4^{2-}]$ measurements in gastight Exetainer© tubes full to the brim and stored at 5 °C. Consecutively, pH was measured using a Hach PHC281101 probe calibrated following the three points procedure (Hach singlet solutions calibrated against NIST standards, precision of ± 0.01 pH unit). Finally, the old water was completely replaced by fresh sterile water.

### 2.1.3 DIC analyses

DIC analyses were performed using 3 mL samples of seawater that were slowly withdrawn from each assay through the Exetainer©
rubber septa using needles syringes. Ultra-pure helium gas was injected in each vial during sampling to ease solution withdrawal
and to prevent atmospheric $CO_2$ contamination. Each 3 mL sample was injected into a new Exetainer© vial, previously flushed
with ultra-pure helium gas (2.5 bar) and loaded with 0.3 mL of 100% $H_3PO_4$. Acidification with pure $H_3PO_4$ converts the total DIC
of the sample into gaseous $CO_2$ which was allowed to degas and mix with the helium gas overnight under shaking. The $CO_2$ and
the He mix was then sampled with an autosampler and sent to a Dual Inlet FinniganTM DeltaPlus XP isotope ratio mass
spectrometer (Thermo Fisher Scientific) (reproducibility = ±0.05‰) at IPGP, Paris. [DIC] was quantified using the linear
relationship between DIC concentration and intensity of the m/z 44 ($^{12}C^{16}O^{16}O$) peak provided by the mass spectrometer (Assayag
et al., 2006). This linear relationship was established based on repeated analyses of internal laboratory carbonate standards
calibrated against international standards (100% calcite), run in different aliquots. The reproducibility for [DIC] measurements
was ± 5% of the measured values ($1\sigma$).

## 2.2 Collection and rinsing procedure of the tests for geochemical analyses

At the end of the culture experiments (that varied between 34 and 39 days), all live individuals of the strain For1C1, as specified
above, those still attached to the substrate from each Petri dish were recovered for geochemical analyses.  Individuals from set 1
and set 2, grown under the same conditions (same medium [$SO_4^{2-}$]), were not combined for analysis. They were measured
separately. Each sample typically weighed few milli-grams. The collected tests were rinsed 3 times in MQ water (basified to pH
9.5 with $NH_4OH$) to remove all traces of salts without dissolving the carbonate phase. In order to remove fresh organic matter,
foraminifera were cleaned following Paris et al. (2014): foraminifera were bathed in a NaOH (0.5M) + $H_2O_2$ (15%) solution at
60°C for 30 minutes. They were then rinsed three times in basified MQ water, and dried overnight at 50°C in a drying oven. All
samples were then dissolved at CRPG (Nancy) in 0.5 ml of 1 to 2% HCl. In addition, in order to determine how the remaining
traces of organic matter could affect $\delta^{34}S$ measurements, some individuals from both For1C1 and C1Tg strains were dissolved in
Aqua Regia (a 50/50 mix of concentrated $HNO_3$ and HCl), without prior cleaning in NaOH and $H_2O_2$. They were left overnight at
120°C and dried down. All acids were distilled at CRPG and the 18.2 M$\Omega$ water purified through a Helga device (Veolia).

## 2.3 Geochemical analyses

### 2.3.1 CAS concentration analysis

In order to determine the $SO_4^{2-}/Ca^{2+}$ ratio of the tests, two dissolved foraminiferal calcite aliquots of 50 µl were used to
independently measure the sulfate and calcium concentrations of the samples. To measure [$SO_4^{2-}$], one of the 50 µl aliquots was
diluted in 200 µl of 18.2 M$\Omega$ water and ran on a Metrohm ion chromatography system (ICS). The calcium content of the samples
was measured using a X-series II ICP-MS using the second aliquot that was dried down and taken up in 3 ml of 2% $HNO_3$. For the
latter, data were measured in groups of 5 bracketed by a 5.3 ppm Ca standard solution and bracketed assuming linear drift between
two standards. In both cases, a calibration line was established to convert the signal to concentrations using home-made
concentration standards. The typical reproducibility for sulfate and calcium concentrations is better than 2% based on multiple
measurements of a diluted seawater solution for sulfate and of the standard solution for calcium.

### 2.3.2 $\delta^{34}S$ analysis

Sulfate isolation from the carbonate matrix was performed by ionic chromatography using the anionic resin Biorad AG1X8 (Paris et al., 2014) using precleaned disposable Biorad columns. Each column was prepared by loading 0.6 ml of resin and rinsed with 10% V/V $HNO_3$ (2x10CV – 1 CV = 1 column volume = 0.6 ml), 33% V/V HCL (2x10CV), 0.5N HCl (1x10CV). After introducing the dissolved carbonates sample on the resin, the column was rinsed with ultrapure water (5x5CV) to remove cations. $SO_4^{2-}$ was then eluted with 0.45 M $HNO_3$ (3x2CV). Each batch of columns included a sample of 50 µl of seawater as a reference and total procedure blanks. After elution, samples were dried down on hotplates with open lids (105°C). The total procedure blank was measured at 0.12 nmol S ± 47 % RSD with a $\delta^{34}S$ value of 6.1 ± 3.5 ‰ 1σ, while the analysed samples contained between at least 23 nmol S (For1C1 grown in ASW[60]). Overall, a blank correction modifies values only within error bars and is thus not applied here.

Purified samples were analysed on the ThermoScientific Neptune Plus MC-ICP-MS at the CRPG using a standard-sample bracketing method (Paris et al, 2013). Samples were run at high resolution using an Aridus-II desolvating membrane to decrease oxide and hydride interferences. Isotopic ratios were collected at m/z 32 and 34 as 1 block of 50 cycles of 4.2 seconds each. Data were corrected offline for instrumental fractionation, drift and background following Paris et al. 2013. Each sample was measured twice on the Neptune and the value provided is an average of both measured ratios. The bracketing $Na_2SO_4$ solution had been previously calibrated against international standard IAEA S1 and checked against IAEA S2 and S3. Seawater samples ran during each Neptune sessions ensure that the data are not biased. Seawater external replicates were measured in association with those samples. They yield an average $\delta^{34}S$ value of 21.1 ± 0.2 ‰ (n=4), in full agreement with published values (e.g. Paris et al., 2013; Present et al., 2015; Rennie et al., 2018). Because carbonate samples were too small to measure full external replicates, we assume the reproducibility for all $\delta^{34}S$ measurements to be the same as seawater (± 0.2‰; 2σ), with the exception of For1C1 grown in ASW [40] and [60]. In these two cases, the reproducibility is calculated based on the weighted mean of the internal errors multiplied by the standard deviation of the External Normalized Deviates (Paris et al., 2013), yielding a 2σ smaller than 0.2 ‰ except for these two samples (0.25 and 0.35 ‰ respectively).

## 3 Results

### 3.1 Population growth in each medium

Individuals For1C1 are morphologically similar to *Rosalina* (Fig. 2). They reproduced asexually when their tests reached a development of 11-12 chambers (Fig. 3 and Fig. 4). Under standard culture conditions of low cell density (i.e. where cells do not compete for food), which in our case was less than about 300 individuals per Petri dish, the reproductive cycle lasted ~ 12 to 15 days and individuals "died" after asexual reproduction by dividing themselves, usually into 20 to 40 viable juveniles, leaving an empty test (Fig. 3). Adult specimens were smaller than the traditional foraminiferal fraction obtained from sieving (through >125 µm mesh) in geochemical studies, they thus may well be common and rarely collected because of their size. A morphological and taxonomic description of the cultured strains is available in the Appendix A. The weekly number of accumulated live individuals incremented by reproduction is given in Fig. 5 and Table 1.

The number of accumulated individuals can trace the population size dynamics for each medium and depends on reproduction rate, number of juveniles produced by individual and mortality. However, while the increase in the number of individuals clearly shows that living cells are being produced, no certainty about their viability can be drawn when the number of individuals stagnates or decreases, as no vital staining has been performed. It was therefore not always easy to distinguish between inactive and dead cells.

We inferred mortality of foraminifera still adhering to the petri dishes from the cessation of reticulopodial activity and cytoplasmic streaming, as well as from the change in cell color (Bernhard, 2000). In the media with no sulfate or sulfate concentrations above 60 mM, we observed little to no reproduction, cell inactivity and probably mortality. As a result, the number of attached foraminifera remained constant and/or decreased over time (Fig. 5, Table 1). The most dramatic reactions were observed within a few hours in the media with highest $[SO_4^{2-}]$ (ASW[120] and ASW[180]), where individuals did not reproduce nor even show any reticulopodial activity. In ASW[90] and ASW[1], only one reproduction cycle was observed and after few days all the cells were inactive (Fig. 5, Table 1). Overall, the highest numbers of individuals at the end of the experiment were obtained in the ASW[28], NSW (Banyuls), ASW[5], ASW[10] and ASW[35] media (Fig. 5). Two media configurations, ASW[10] and ASW[28], from set 1 experiments were replicated in set 2 experiments. If the abundances for condition ASW[10] are of the same order of magnitude in both sets, the abundances for condition ASW[28] are much lower in set 2 compared to set 1. This was related to the reproduction rate in set 2, which slowed down drastically after 15 days. This decrease can be explained by a microbial bloom in the media that was observed in no other media (Appendix, Fig. C1). The microbial spread could not be reduced by the weekly water change, and any transfer and rinsing of foraminifera or antibiotic treatment would have constituted an additional experimental modification. We thus kept counting foraminifera and sampling seawater, but did not take into account pH and DIC value nor foraminifera counts measured in that media after day 15.

**3.2 pH and DIC evolution**

pH variations remained within ± 0.3 pH units during each experiment (Table 2).

pH drifted from the starting point between 8.1 and 8.2 towards more acidic values (7.83 minimum) and was reset close to 8.2 at each medium change for the first 15 days and then remained rather stable with values varying between 8.19 and 8.07. DIC ranged from 3.2±0.2 mM (2σ) to 4.2±0.3 mM (2σ) (Table 3). These concentrations are higher than the theoretical initial concentration of 2.8 mM using the recipe of Kester et al. 1967. While in Kester et al.'s recipe, the targeted 8.2 pH is achieved after 2h equilibration with the $CO_2$ in the atmosphere, we had to proceed to NaOH addition despite a similar equilibration time. It is possible that higher $CO_2$ dissolution at the atmospheric pressure of the year we performed the experiments (407 ppm in the atmosphere and probably more in the lab against 322 ppm in 1967), led to an increase in DIC. In addition, DIC probably built up in the Petri dishes each week as the foraminifera respire. pH and DIC variations for cultures in ASW[28] and ASW[10] are shown in Fig. 6.

**3.3 CAS concentration**

CAS concentration in foraminiferal calcite was performed for the media ASW[5], ASW[10], ASW[28], ASW[35], ASW[40] and ASW[60], as the other samples were lost during the manipulations or were below the detection limits. The obtained values are presented in Fig. 7 and Table 4. Each datapoint was obtained using hundred to several hundreds of foraminifera for each medium. CAS concentration (sulfate to calcite ratio) increased from 3320 ppm to ~14000 ppm $SO_4^{-2}/CaCO_3$ (±5%, 2σ) in proportion to total $SO_4^{2-}$ concentrations in artificial seawater which increased from 5 mM to 28 mM. Above the modern seawater concentration of 28 mM, the foraminiferal CAS concentration no longer covaries with seawater and remains on a plateau between seawater $[SO_4^{2-}]$ 28 mM and 60 mM (Fig. 7). This suggests that a threshold (~14000 ppm) is probably reached at about 28 mM $[SO_4^{2-}]$ in seawater. Because we only have a natural seawater replicate for the sulfate concentration of modern seawater (28 mM), we observe a scatter at 28 mM that makes it difficult to determine precisely when the plateau starts. The slight decrease in foraminiferal CAS to 9740

ppm at ASW[60] is actually part of the variability of CAS values at 28 mM [$SO_4^{2-}$] in seawater (Fig. 7 and Table 4). The foraminiferal CAS values from the ASW60 configuration can therefore be considered as part of a plateau (Fig. 7).


### 3.4 Sulfur isotopic composition

The $\delta^{34}S$ values of the foraminiferal CAS from the different media are plotted in Fig. 8 and listed in Table 4. Measurements were performed both on foraminiferal samples from the C1Tg and For1C1strains cultured in NSW or ASW[28] during the acclimation
period and in those coming from the [$SO_4^{2-}$] variation experiment (For1C1strain only), from a selection of culture media in which [$SO_4^{2-}$] varied from 5 to 60 mM (Table 4). NSW $\delta^{34}S$ composition was measured before (21.1±0.2‰) and 7 days after adding the algae (19.9±0.2‰). There was a difference beyond error bars between the two values. Considering that algae $\delta^{34}S$ composition are of 7.0±0.2‰ the difference may be explained by the isotopically depleted sulfate added resulting of algae decomposition, lowering the average $\delta^{34}S$ of the media. This effect was not detectable in ASW possibly because the $\delta^{34}S$ values of medium (9.1±0.2‰ for
ASW[28] and 0.1±0.2‰ for ASW[5], ASW[10], ASW[35], ASW[40] and ASW[60]) were closer to that of the algae (Appendix B Table B1). Considering that algae were added at each water change and degraded within 1 or 2 days, and that foraminifera entered into a chamber formation sequence after feeding (Fig. 4), we consider that the seawater $\delta^{34}S$ that prevailed during chamber formation is the value measured after several days of culture with algae, 19.9±0.2‰ in NSW and 9.1±0.2‰ or 0.1±0.2‰ in ASW[28] (Appendix B Table B1). A $\delta^{34}S_{CAS}$ - $\delta^{34}S_{sw}$ fractionation value of 1.6 ±0.3‰ (as observed for For1C1 pool (1 sd, 8
samples in total coming from all [$SO_4^{2-}$] concentrations) while it was 1.4±0.2‰ for C1Tg specimens (1sd, 5 samples in total coming from NSW or ASW[28]), which is indistinguishable within the error range (Fig. 8).

Samples for which organic matter was preserved yielded $\delta^{34}S$ values of 1.1±0.2‰ (For1C1 in ASW[28]) 0.4±0.2‰ (For1C1 in NSW) 1.4±0.2 ‰ (C1Tg in NSW) and 0.5±0.2‰ (C1Tg in ASW[28]) lower than the value that was obtained for the For1C1 and C1Tg tests from which organic matter had been oxidatively removed (Table 4).


### 4 Discussion

### 4.1 [$SO_4^{2-}$] changes in seawater can affect foraminiferal biology

Our results highlight that a change in seawater [$SO_4^{2-}$] concentration can affect foraminiferal cellular activity, reproduction and population size.

Reticulopodial activity stopped few hours after the transfer of individuals of the For1C1 strain from 28 mM of sulfate (ASW[28]) to concentrations above 120 mM (ASW[120] and ASW[180]) or without sulfate (ASW[0]). Dissolved sulfate and food were the only sources of sulfur in this experiment, which is essential for life. Since ASW[0] prevented any reproduction and induced cellular
inactivity, we infer that sulfur from food appears insufficient and that dissolved sulfate in seawater is necessary for cellular activity in foraminifera. At the other extreme, toxic impact of the highest [$SO_4^{2-}$] (ASW[120] and ASW[180]) can explain the non-reproduction and the cellular inactivity of individuals after a few hours. For1C1 individuals survived and even reproduced once in the ASW[1] and ASW[90] media (Fig. 5). Thus, our results suggest that foraminifera can reproduce and show pseudopodial activity only within a certain range of [$SO_4^{2-}$], from 1 to 90 mM, extreme values at which the cellular activity is already very low.
Individuals appear to tolerate these extreme conditions for only the first week and then cease all reproductive activity. They appear to be adapted, beyond the modern oceanic [$SO_4^{2-}$] (28.2 mM) to a range of seawater [$SO_4^{2-}$] from 5 to 35 mM, as shown by the high number of accumulated live individuals incremented by reproduction at the end of set 1 and set 2 experiments (Fig. 5). As

already mentioned, the low number of individuals at the end of the second set of experiments is due to a bacterial proliferation after 3 weeks. Population size are growing less fast above $[SO_4^{2-}]$ of 35 mM and below 5 mM, suggesting a foraminiferal reproduction sensitivity to $[SO_4^2]$ variations.

The effect of changes in seawater $[SO_4^{2-}]$ on foraminiferal reproduction highlights a possible mechanism by which changes in the composition of seawater can affect the carbonate record. It has previously been hypothesized that seawater Mg/Ca ratio and $SO_4^2$/Ca ratio, control the switch between calcite and aragonite dominance in the sedimentary record, as a high $SO_4^{2-}$ and Mg concentrations in seawater inhibit calcite precipitation and promote aragonite precipitation, as shown by inorganic precipitation experiments (Bots et al., 2011; Barkan et al., 2020). Here, we show that a change in seawater $[SO_4^{2-}]$ may also affect foraminiferal reproduction, population size and hence their calcitic test accumulation in the sediment. However, this appears to be for seawater $[SO_4^{2-}]$ variations far below and above the range thought to be involved in long-term secular variations in the Phanerozoic (~5-28 mM), suggesting an adaptation of foraminifera in this range of variations. Indeed, under conditions that mimic the Phanerozoic range of $[SO_4^{2-}]$ variations, reproduction and population growth appear to be unaffected.

## 4.2 Foraminifer CAS concentration versus seawater $[SO_4^{2-}]$ or $S/CO_3^{2-}$

Our cultured foraminifera contain high levels of sulfates, similar to high Mg-calcite foraminifera previously grown during culture experiments, and significantly higher than low Mg-calcite foraminifera (Paris et al., 2014; van Dijk et al., 2017; 2019) (Fig. 7, Appendix B, Table B2). Similar to previous results of foraminiferal culture experiments comparing CAS content with seawater sulfate concentration, we note an increase in foraminiferal CAS content with seawater sulfate concentration increase. More specifically, our results show that CAS concentration in foraminiferal calcite grown in experimental seawater increases with seawater $[SO_4^{2-}]$ concentration from 5 to 28 mM (Fig. 7), similarly to what is observed in inorganic carbonates (Busenberg and Plummer, 1985; Fernandez-Diaz et al., 2010; Barkan et al., 2020) or previous foraminiferal investigation (Paris et al., 2014). At $[SO_4^{2-}]$ higher than 28 mM in seawater, the incorporation of sulfate in the foraminiferal calcite seems to reach a saturation point (Fig. 7). It is remarkable to note that foraminifera can reproduce and thus calcify at $[SO_4^{2-}]$ as high as 90 mM (Fig. 5), concentrations at which no inorganic calcite precipitation occurs (Bots et al., 2011; Barkan et al., 2020). However as discussed before, their reproduction is limited to the first week, which strongly suggests that they could only tolerate brief exposure to such a high level of sulfates in their environment.

A geochemical modeling of experiments in which CAS, pH and DIC was measured is available in Appendix D, and permitted us to extract $CO_3^{2-}$ concentrations. Overall, we observe an increase and a plateau, whether we compare our CAS content to seawater sulfate concentration or $S/CO_3^{2-}$ ratios. When we replace total sulfate (the sum of free $SO_4^{2-}$ and its major complexed forms (NaSO_4, CaSO_4 and MgSO_4) by only free sulfate, the linearity of the 5mM to 28mM CAS accumulation trend is maintained. However, the plateau from 28 to 60 mM is less visible, potentially evidencing the role of complexes formation in the lower $SO_4^{2-}$ incorporation in the tests. As shown in the appendix, the sole formation of complexes cannot explain the plateau observed in figure 7.

To understand this evolution of sulfate content, we must first describe where sulfur is located in the test. Two options are possible:

i) CAS: Sulfate is incorporated into both inorganic and biogenic $CaCO_3$ minerals as CAS within the growing mineral structure, the larger tetrahedral sulfate substituting to the smaller trigonal-planar carbonate ion (Busenberg and Plummer, 1985; Kontrec et al. 2004; Balan et al., 2014; Tamenori et al. 2014; Perrin et al. 2017; Tamenori and Yoshimura 2018).

ii)$S_{org}$: Sulfur present in the organic matrix used by biomineralizing organisms to initiate calcification and orient the growing crystals (e.g. Cuif et al., 2004; Richardson et al., 2019; de Noojier et al., 2014). The organic matrix of the test of a wide variety of foraminiferal taxa contains over-sulfated glycosaminoglycans and proteins (Weiner and Erez,1984; Langer 1992). The benthic foraminifera Rosalinidae belong to the order Rotaliida and likely share the same mechanisms of biomineralisation and bilayer test construction as other rotaliid families. In the case of rotaliid test, the calcareous wall growth of each new chamber results from the bioprecipitation of two calcite layers, on either side of an organic matrix (Bé et al., 1979; de Noojier et al., 2014; Nagai et al., 2018) referred to as the Primary Organic Sheet (POS, Erez, 2003). However, since we applied an oxidative cleaning to the foraminiferal tests to destroy the organic matter, we assume that most of the measured $[SO_4^{2-}]$ in the tests are linked to the CAS concentration, although a small contribution might be still associated with $S_{org}$ within the biomineralized calcite (Burdett et al., 1989; Cuif et al., 2003; Paris et al. 2014). In the following discussion, we will thus assume that our measured sulfate content reflects structurally-bound CAS.

Several hypotheses can be formulated to explain this $SO_4^{-2}/CaCO_3$ incorporation pattern:

i) Foraminifera may be able to regulate $[SO_4^{2-}]$ at the site of calcification (SOC) during calcite precipitation through active transmembrane transport, removing excess sulfate and lowering it in the precipitating fluid, enabling calcite nucleation and precipitation, as sulfate in high concentration inhibits calcite precipitation and makes it more soluble (Busenberg and Plummer, 1985; Bots et al., 2011; Barkan et al., 2020). In fact, under our experimental conditions the amount of CAS incorporated in foraminiferal calcite correlates with seawater $SO_4^{2-}$ concentration, from 5 up to a plateau that starts at 28 mM. The mere fact that calcite precipitates therefore suggests that sulfate is at least partially removed from the precipitating fluid, altering the local $SO_4^{2-}$ concentration. The correlation suggests that this removal is partial and, to some extent, proportional to the concentration of $SO_4^{2-}$ in seawater.

ii) An increase in the carbonate ion concentration may help maintaining a constant $SO_4^{2-}/CO_3^{2-}$. Previous investigations demonstrated that it is more appropriated to reason in terms of $SO_4^{2-}/CO_3^{2-}$ ratio of the calcifying fluid rather than $[SO_4^{2-}]$ as sulfate substitutes for $CO_3^{2-}$ in the forming mineral (van Dijk et al., 2019; Barkan et al., 2020). Another way to maintain $SO_4^{2-}/CO_3^{2-}$ constant while $[SO_4^{2-}]$ increases would be to proportionally increase $CO_3^{2-}$. Like other calcifying organisms, benthic foraminifera modify the pH of the precipitating fluid to promote calcite formation (Erez, 2003; de Noojier et al., 2008; Rollion-Bard and Erez, 2010; Toyofuku et al., 2017). Foraminifera most probably actively pump protons out of the SOC (Sabbatini et al., 2014; Toyofuku et al., 2017). Intensifying this process in case of elevated $[SO_4^{2-}]$ would induce an increase in carbonate ion concentration (and the saturation state) and could therefore help to maintain a constant $SO_4^{2-}/CO_3^{2-}$ when $[SO_4^{2-}]$ reaches values between 28 and 90 mM, allowing calcite bio-precipitation. This mechanism, like that of active sulfate transmembrane transport mentioned above, would cease to function at seawater $[SO_4^{2-}]$ levels neighboring 120 mM, when the foraminiferal cells become inactive.

iii) A preferential sequestering of sulfate in some organic rich layers at the incipient phase of biocalcification might allow to decrease the $[SO_4^{2-}]$ in the remaining liquid and thereby prevent further sulfate incorporation into foraminiferal calcite above 28 mM $[SO_4^{2-}]$ in seawater. High resolution sulfur nano-mapping on transversal section of perforate foraminiferal tests (such as Rosalinidae or *Orbulina*) showed a banded heterogeneity in sulfur distribution across the multi-layer structure (Paris et al., 2014; van Dijk et al., 2019). XRF intra-test mapping revealed a preferential incorporation of metals and sulfur in the POS zone, the organic incipient stage of the build-up of the wall of a new chamber of test (Lemelle et al., 2020). In our case, organic matter has been oxidized, and most of the "stored" $SO_4^{2-}$ was likely removed.

iv) A kinetic effect could also explain the non-linearity of the CAS concentration in foraminiferal tests with corresponding increases in $[SO_4^{2-}]$ above 28 mM, as inorganic calcite precipitation experiments suggest a reduction in crystal growth rates at higher $[SO_4^{2-}]$ (Busenberg and Plummer, 1985). However, it is worth noting that a decrease in precipitation rate can also be associated to a lower CAS content in inorganic calcite (Barkan et al., 2020). As a result, one could imagine that the change in sulfate concentration reflects a change in precipitation rate induced by different sulfate concentration in seawater and/or in the biomineralizing fluid. However, as calcite is more soluble and precipitates less easily at high sulfate concentration, we would expect an opposite effect to what we observe in the 5-40 mM part of our results. There could nonetheless be a contribution of the precipitation rate effect to the plateau we observe.

v) Finally, another possibility to explain why the CAS in foraminiferal tests does not increase linearly with corresponding increase in the $[SO_4^{2-}]$ beyond 28 Mm, could be that at such concentrations in solution, sulfate might complex more easily with other cations such as $Ca^{2+}$, $K^+$, $Mg^{2+}$, $Na^+$, $Sr^{2+}$, etc. (Garrels and Thompson, 1962). Such complexes cannot be effectively incorporated into the mineral lattice structure. This might influence the amount of $SO_4^{2-}$ substituted in carbonates, and thus the CAS in foraminiferal tests. A geochemical model, available in the Appendix D, taking into account our media configurations where we had CAS, pH and DIC data, shows that the model CAS concentration follows the seawater $SO_4^{2-}/CO_3^{2-}$ and $CaSO_4/CaHCO_3$ concentrations, which in turn depend mainly on the $[SO_4^{2-}]$ in solution, with a slowed increase between 40 and 60 mM, likely related to the formation of complexes. However the sole formation of complexes cannot explain the plateau observed beyond 28 mM.

The putative mechanisms for sulfate regulation could have been adopted by foraminifera as evolutionary strategies to maintain carbonate precipitation despite potential variation in $[SO_4^{2-}]$. Indeed, at $[SO_4^{2-}]$ greater than 8 mM abiotic calcite nucleation and precipitation is inhibited, and aragonite precipitates from saturated solutions (Kitano and Hood, 1962; Kitano et al., 1975; Bots et al., 2011). This inhibition is also true in the lack in magnesium (Barkan et al., 2020) and thus sulfate alone can affect calcite precipitation. Mechanisms such as increasing calcium concentration, pH and/or saturation state (e.g. Zeebe and Sanyal, 2002; Nehrke et al., 2013; Evans et al., 2018), as well as the presence of organics, could help overcome such high concentration of sulfate. However, when it comes to magnesium, active removal is also an option (Bentov and Erez, 2006). Calcitic foraminifera, which first appeared during the Devonian (Vachard et al., 2010) in a range of low-sulfate seawaters of ~3-15 mM (Algeo et al., 2015), might have progressively adopted such strategies in order to precipitate calcite in high-sulfate (~28 mM) seawaters such as those in the present-day ocean, and preserve the capacity to precipitate calcite at concentrations reaching 90mM as evidenced here. However, in addition to active biological control to remove sulfate from the calcification site, a reduction in sulfate uptake in the tests at high seawater sulfate concentrations (> 28 mM) is likely to be due to the formation of sulfate complexes with other cations, explaining also the non-linearity of the CAS concentration in foraminiferal tests at high increases in $[SO_4^{2-}]$.

**4.3 Sulfur isotope fractionation**

The isotopic composition of CAS remains constants through our experiments. Sulfur isotopic fractionation of CAS in benthic foraminifera (Rosalinidae) is not sensitive to the variation in $[SO_4^{2-}]$ in seawater (Fig. 8), thus confirming the earlier observation on planktic foraminifera by Paris et al., 2014. This result by itself is important and confirms that foraminiferal CAS constitutes a reliable proxy of seawater $\delta^{34}S$. This result, together with the correlation between $SO_4/CaCO_3$ and seawater $[SO_4^{2-}]$ (Fig. 7), supports that CAS in foraminiferal tests is of inorganic origin.

More importantly, the fractionation observed here is clearly different from the inorganic fractionation measured in the inorganic calcite (Barkan et al., 2020) highlighting the involvement of some biological isotopic fractionation. Considering that the algae's

organic sulfur source had a fixed sulfur composition (7 ‰) while the seawater $\delta^{34}S$ varied from one medium to the other (from -0.1 to 20.0‰), our isotopic measurements on $S_{org}$+CAS allow to infer the origin of $S_{org}$ as well. Mass balance calculation permit to determine that the isotopic composition of $S_{org}$ varies with seawater $\delta^{34}S$ value, pointing towards mainly an inorganic source for $S_{org}$ (Fig. 8). This is consistent with our observation that no cellular activity of foraminifera was possible in medium with zero $[SO_4^{2-}]$, even in the presence of algae as food and possible source of $S_{org}$. The $\delta^{34}S$ value of the combined S pool ($S_{org}$ + CAS) is 0.4 to 1.4 ‰ more negative than the $\delta^{34}S$ value of CAS alone, which points towards the involvement of some biological fractionation or vital effect associated to the incorporation of sulfur.

**4.4 Implication for Paleoenvironmental reconstructions**

Sulfur isotopic composition in the sedimentary record, through sulfur species redox reactivity and multiple deposition form, records several paleoenvironmental processes occurring in the atmosphere and the ocean (Farquhar et al., 2000; Farquhar and Wing, 2003; Crockford et al., 2019). This makes sulfur one of the most studied elements for the surface processes. And yet, in order to investigate the sulfur cycle, it is necessary to interrogate different sedimentary archives: carbonates, barites, evaporites as well as pyrites (Paytan et al., 1998; Algeo et al., 2015; Halevy et al., 2012; Present et al., 2020) to reconstruct variations in both $\delta^{34}S$ and $[SO_4^{2-}]$. The work to match the sedimentary record of both $\delta^{34}S$ and $[SO_4^{2-}]$ is laborious and requires calibrations. Our results show that benthic foraminifera (Rosalinidae) incorporate CAS in their test proportionally to the $[SO_4^{2-}]$ in seawater, at least in the 5-28 mmol/L range, confirming previous experiments on planktic foraminifers that foraminiferal CAS can serve as a proxy for variations of both $\delta^{34}S_{CAS}$ and $[SO_4^{2-}]$ in seawater (Paris et al., 2014). However, they also highlight that above the seawater $[SO_4^{2-}]$ of 28 mM, it might not be possible to confidently determine the seawater $[SO_4^{2-}]$ using foraminiferal CAS, as the previous linear correlation no longer holds. This limitation means that foraminiferal CAS could be used to trace deep time secular changes in seawater $[SO_4^{2-}]$, which varies from about 5 mM to 28 mM today (Algeo et al. 2015), but probably not to trace past seawater $[SO_4^{2-}]$ enrichments above 28 mM, such as those that could be caused by important volcanic eruptions or sulfate-rich volcanic hydrothermal fluids on the seafloor. Future works are therefore important to confirm whether or not the seawater $[SO_4^{2-}]$ threshold of 28 mM for CAS incorporation can be applied to other benthic and planktonic foraminifera, or whether it is restricted to Rosalinidae.

The use of CAS concentration as a marine $[SO_4^{2-}]$ record is still promising, despite the limitation discussed above, but will require calibration on various types of carbonates and species that may each have their own fractionation factor. The preservation of that dual $\delta^{34}S/[SO_4^{2-}]$ in foraminiferal calcite has to be evaluated in the carbonate record, as diagenesis has the capacity to affect $[SO_4^{2}]$ in carbonates (e.g. Gill et al., 2008; Marenco et al., 2008; Rennie and Turchyn, 2014).

Additionally, it has been previously supported that S/Ca can work as a proxy for $CO_3^{2-}$ concentration (van Dijk et al., 2017). Our results complement this finding under the condition that it is applied on timescales where seawater $[SO_4^{2-}]$ are constant.

The other major implication of our results for the interpretation of the geological record is that changes in seawater $[SO_4^{2-}]$, could affect the production of carbonate by affecting the reproduction/survival of at least some calcifying organisms, as the benthic foraminifera studied in this work. In theory, the increase in seawater $[SO_4^{2-}]$ is expected to have a purely "abiotic" effect on calcite production as sulfate thermodynamically inhibits calcite formation and makes calcite more soluble. As a result, higher sulfate content in the living medium would generate a decrease in calcification intensity for a given individual. In this experiment we showed that $[SO_4^{2-}]$ higher in the medium than those of the modern ocean can also decrease the amount of accumulated calcite by affecting foraminiferal population size, suggesting that their biological activity is harmed by such sulfate concentrations. As a result, the decrease the total amount of calcification would be explained partly by a decrease in biological activity. This work illustrated how variations in seawater composition can have a dual effect on biomineralizing organisms. Conditions that inhibit

calcite formation such as increases in marine concentrations of $Mg^{2+}$ or $SO_4^{2-}$, could have chemical "abiotic" effects on carbonates formation but could also affect biological processes involved in biomineralization.


## 5 Conclusion

We cultured rotaliid foraminifera in media with $[SO_4^{2-}]$ spanning from 0 mM to 180 mM, stable salinity and fixed seawater $\delta^{34}S$.

$[SO_4^{2-}]$ changes in seawater affected foraminiferal reproduction, population size and hence test accumulation. Foraminifera kept precipitating calcite in media reaching $[SO_4^{2-}]$ = 90 mM but very temporarily. Sulfate from seawater is necessary for the cellular activity of foraminifera, but at concentrations equal and above 90 mM it becomes toxic to them, as evidenced by cellular inactivity and reproductive arrest. Sulfur concentration in CAS varied proportionally to seawater $[SO_4^{2-}]$ between 5 and 28 mM and then stabilizes. Our results highlight that isotope fractionation between CAS and seawater does not depend on seawater $[SO_4^{2-}]$. Overall,

similarly to planktic foraminifera the $\delta^{34}S_{CAS}$ value of a given species of benthic foraminifera is a reliable way to reconstruct seawater $\delta^{34}S$, despite variations of $[SO_4^{2-}]$ in seawater.

**Author contribution:** All authors participated in designing and interpreting the experiments. CT carried out culture experiments under MD and AB supervision and technical support from AL. MD isolated the strains and developed the cultivation protocol.

GP performed all isotopic and $SO_4/CaCO_3$ analyses. CT designed the figures and wrote the paper with contributions from all co-authors.

**Competing interests:**

The authors declare no competing interests


**Acknowledgements**

This research was funded by the LabEx BCDiv project « SULFOR, Impact of $SO_2$ on ocean acidification and foraminiferal biocalcification» (PI AB); the CNRS INSU INTERRVIE project « Impact of sulfate variations on the biocalcification of foraminifera » (PI AB). Sylvain Pont (MNHN, IMPMC UMR 7590, France) produced the SEM imaging. The authors thank the

reviewers Julien Richirt and David Evans for their very constructive criticism, which helped to improve the article.

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

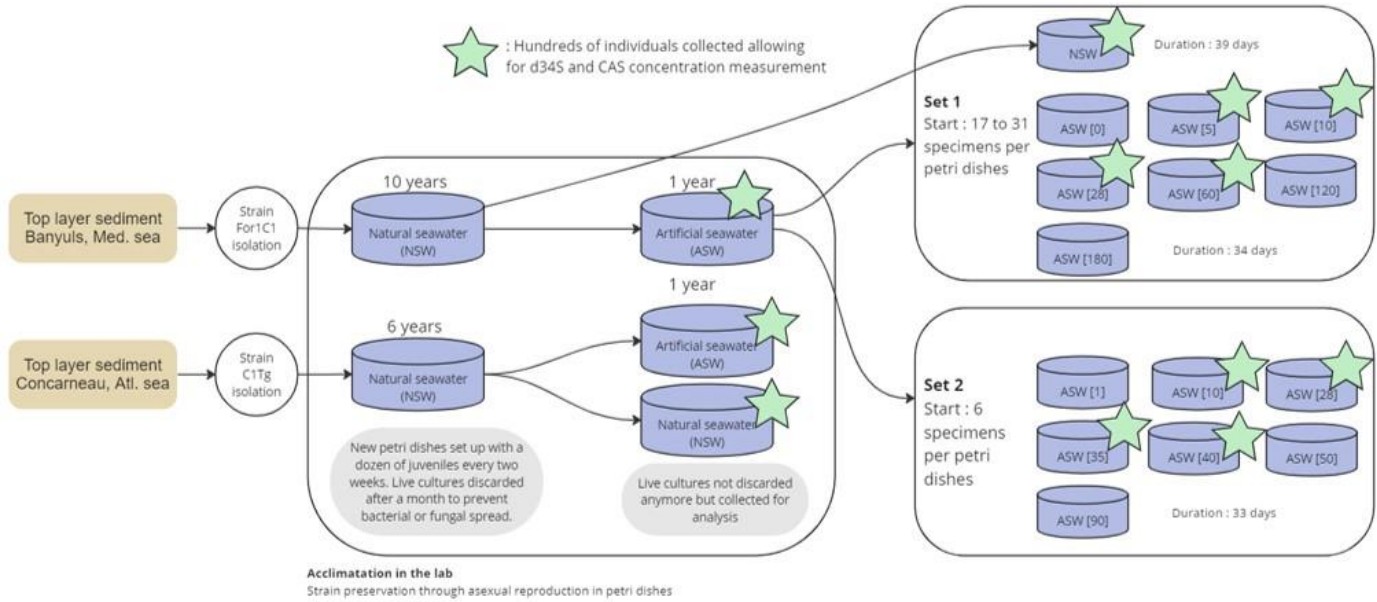

**Figure 1. Experimental design workflow diagram illustrating the sampling, acclimation, and experiments SET 1 and SET 2. Stars highlight samples where δ³⁴S and CAS analysis could be performed.**

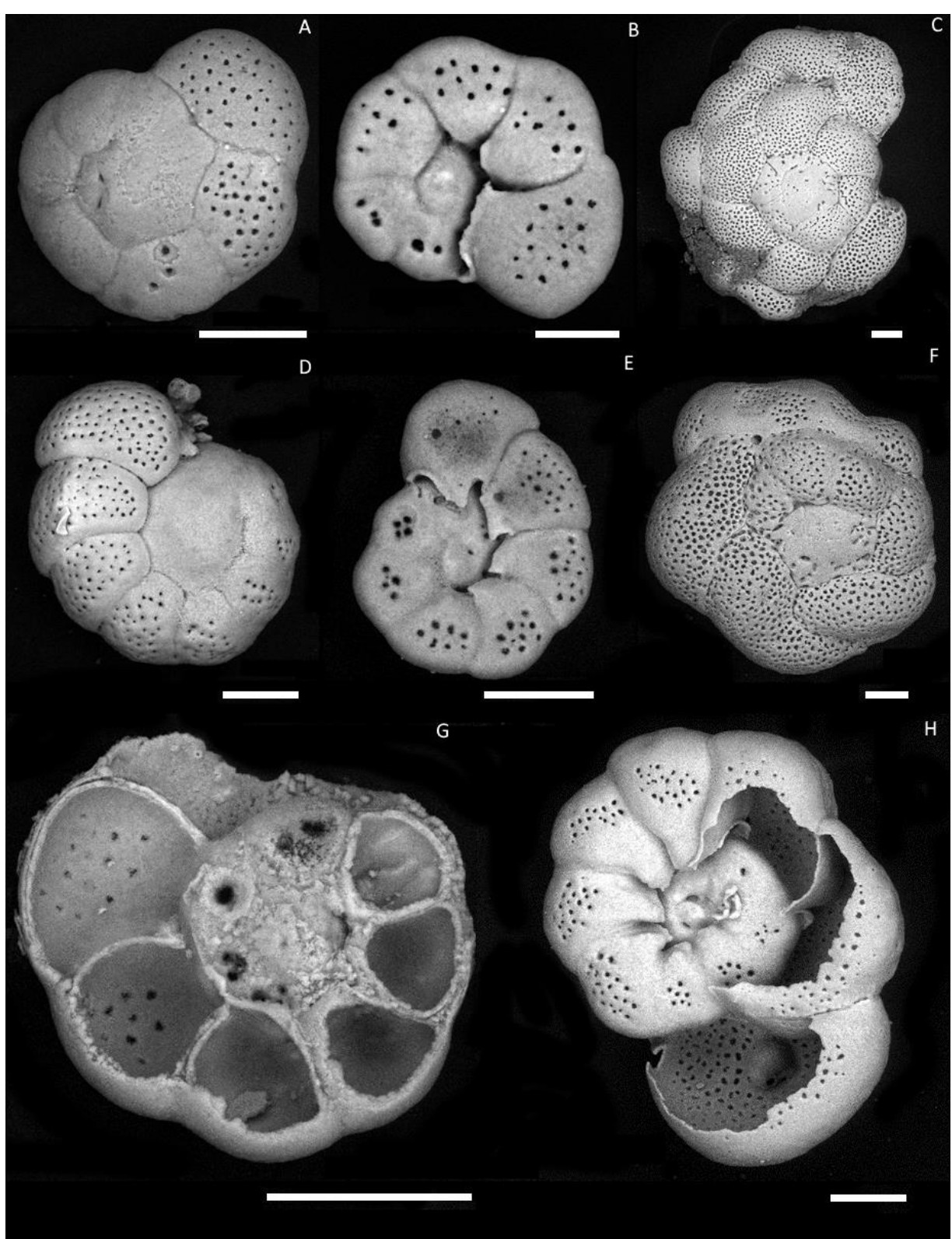

**Figure 2. Foraminiferal strains cultured in this study. For1C1: (A)** *Rosalina* **like morphotype (11-12 chambers) reproducing asexually, dorsal view, (B) same as A, ventral view, (C) Morphotype with more than 12 chambers, starting as** *Rosalina* **morphotype and then developing annular disposition of the last chambers, dorsal view. C1Tg: (D)** *Rosalina* **like morphotype, (E) same as D, ventral view, (F) morphotype with annular arrangement of the last chambers, dorsal view. (G and H) ventral view of C1Tg with a broken test permitting to see the layered structure of the test's wall (G) and the foramen position inside of the test (H). Scale bar 50 µm, SEM picture in BSE**
**mode operated at 10 to 22 mPa and 20 000kV.**

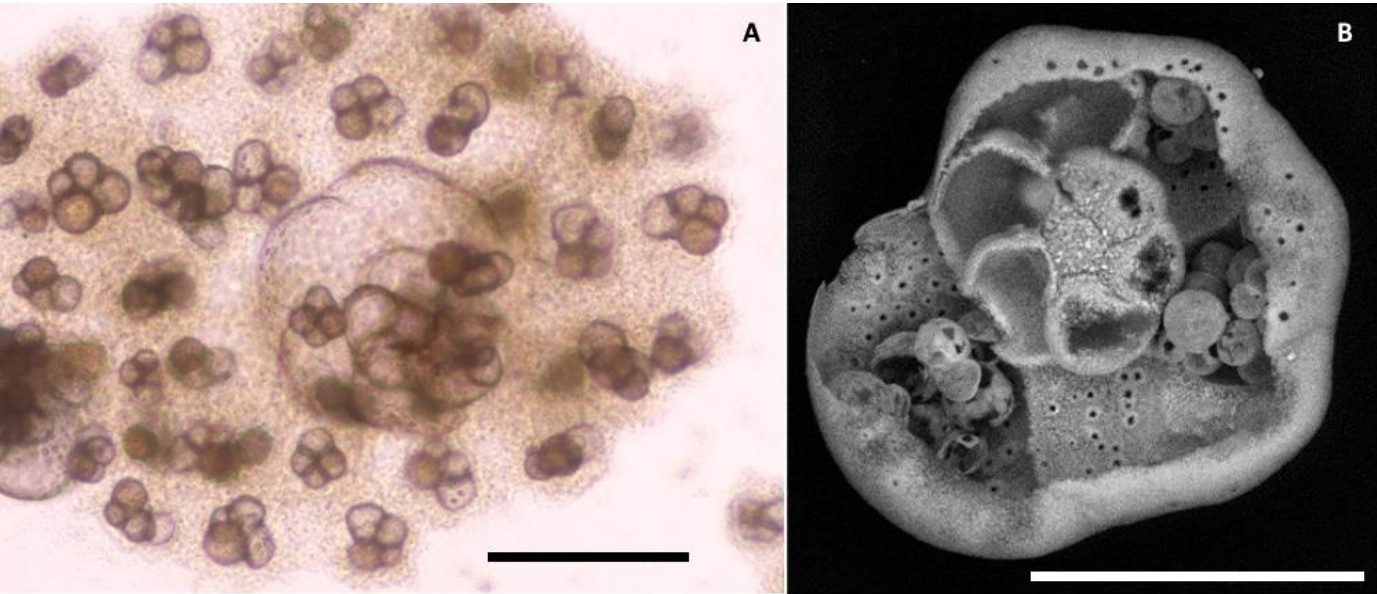

**Figure 3. Asexual reproduction of an individual of the For1C1 strain. (A) Light microscope image of a megalospheric schizont adult that has ~12 visible chambers, and whose cell has divided asexually into viable juveniles (for further detail, see Appendix A). The darker appearance of the juveniles compared to the adult is due to the presence of cellular material. After division, the adult is empty and its test partially dissolved, as shown in the SEM micrograph (B). Scale bar 100 μm.**


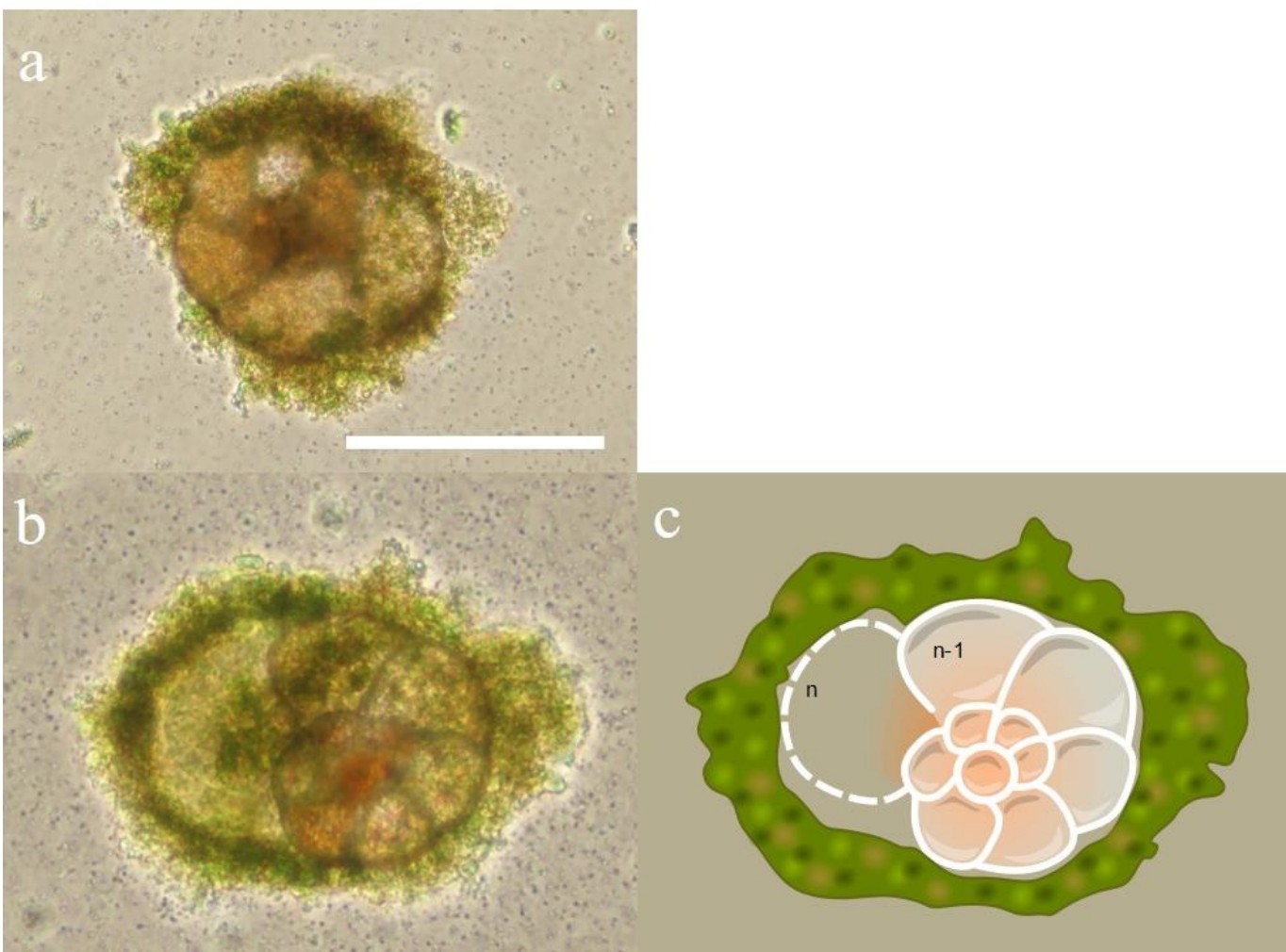

**Figure 4. Chamber formation during day 2 of the set 1 series of experiment. Two For1C1 individuals (from Banyuls) in (a) ASW[5] (containing 5 mM $SO_4^{2-}$) and (b) ASW[10] (containing 10 mM $SO_4^{2-}$) and its schematic representation illustrating the new chamber (n)**

**formation and the surrounding gangue (algal cyst) constituted by the foraminifer by the accumulation of foreign detritus and other materials, confining the new chamber in formation in a microenvironment. In the case of rotaliid foraminifera, the formation of a new chamber begins with the isolation of the chamber volume from the surrounding environment by a structure which probably forms the organic scaffolding that shapes the morphology of the chamber and serves as a template for the calcification of the wall (Bé et al., 1979; De Nooijer et al., 2014; Nagai et al., 2018). Precipitation of calcium carbonate takes place on both sides of an organic layer, called primary**

**organic sheet (POS, Erez, 2003), sandwiched between the outer and inner organic layers. Scale bar 100 µm.**



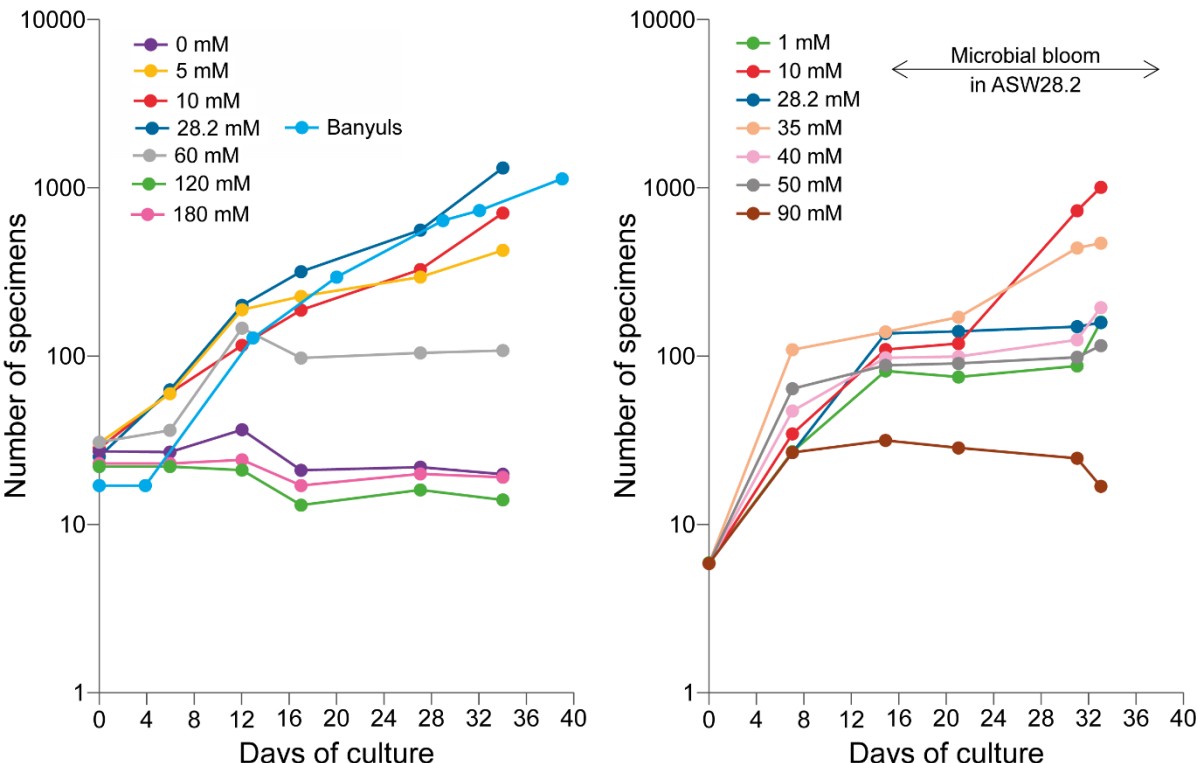

**Figure 5.** Evolution of the number of the attached specimens of the For1C1strain in each culture medium at different [$SO_4^{2-}$] for **(a)** Set 1 and **(b)** Set 2. The number of foraminifera counted (with a precision of ± 3 individual) corresponds to the living ones adhering to the bottom of the Petri dish before the change of medium. The increase in the number of individuals is due to asexual multiplication (see Appendix A). In Set 1, the largest population in terms of size occurs for 28.2 mM (ASW[28] and NSW Banyuls). In Set 2, a microbial bloom occurred after 12 days in medium ASW[28], likely affecting the reproduction rate (Appendix C, Appendix Fig. C1). The y-axes are on log scales.

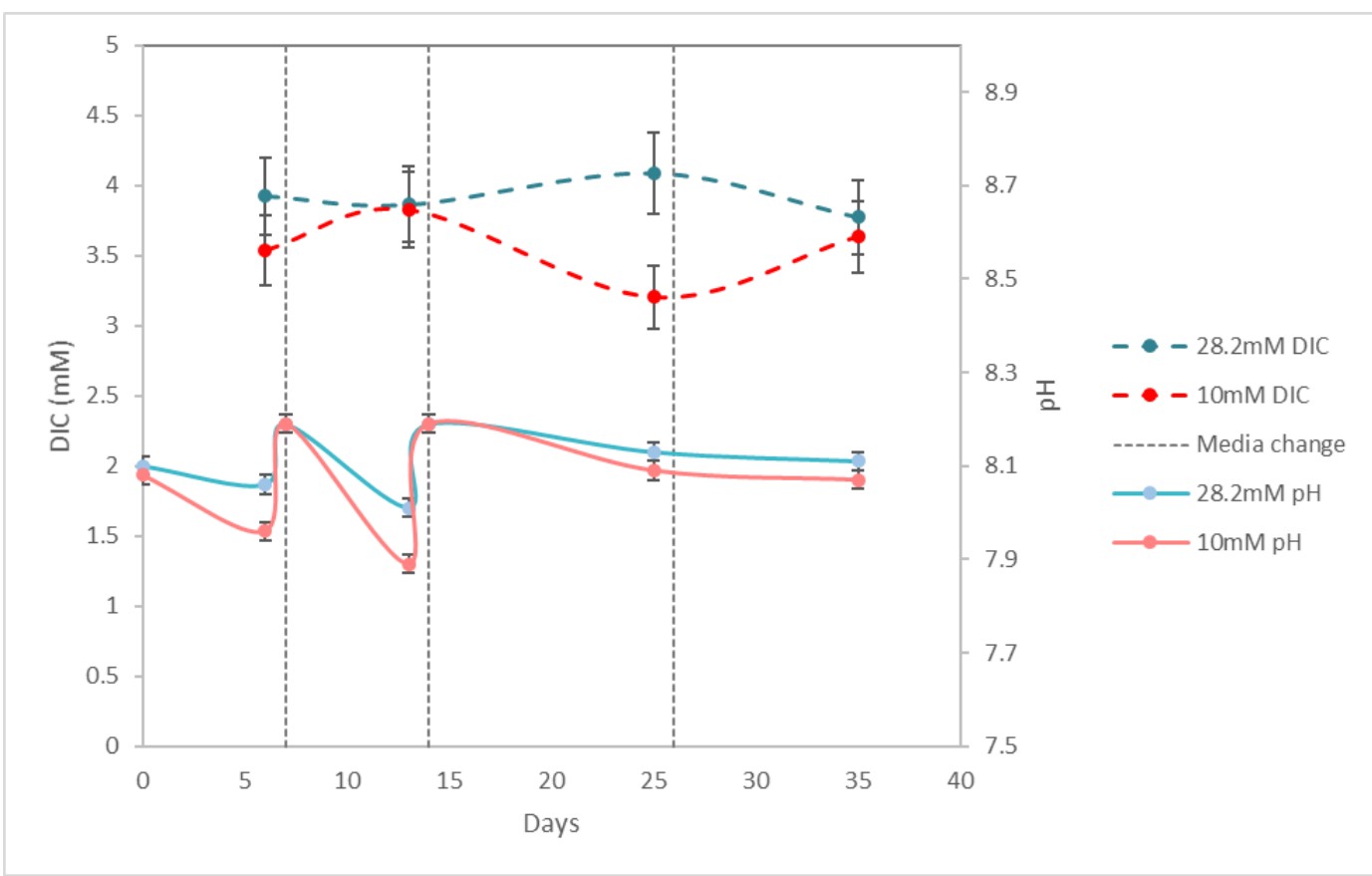

**Figure 6. Evolution of pH and DIC over 35 days of For1c1 culture at 10 mM and 28.2 mM during SET2, with a change of medium on days 7, 14 and 26. Measured DIC in all experiments (Table 3) varies between 3.2 mM and 4.1 mM, it decreases by calcite formation and increases by specimen respiration. The variation of pH in all experiments when measured (Table 2) is between 7.89 and 8.19. The decrease in pH is caused by both respiration ($CO_2$ production) and calcite precipitation.**

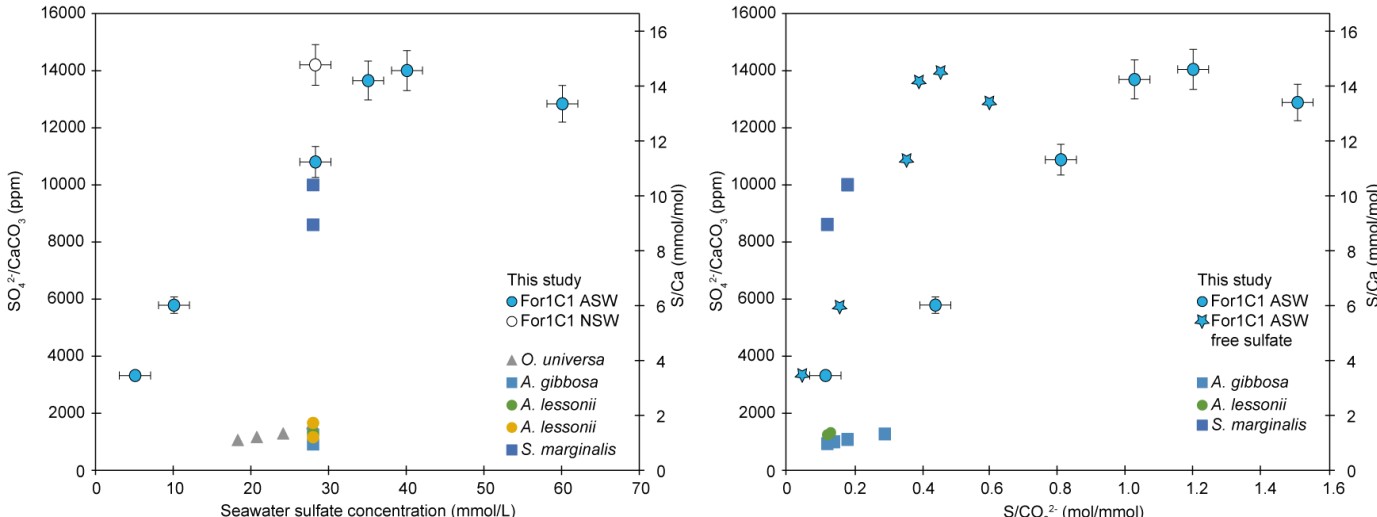

**Figure 7.** Left panel: $SO_4^{2-}/CaCO_3$ and S/Ca ratios on tests of the For1C1 strain at the end of SET1 and SET2 experiments, as a function of seawater $[SO_4^{2-}]$ (5, 10, 28.2, 35, 40 and 60 mM). Right panel: $SO_4^{2-}/CaCO_3$ and S/Ca ratios on tests of the For1C1 strain at the end of SET1 and SET2 experiments, as a function of seawater $S/CO_3^{2-}$ . For our experimental results, we report the values using both S as the sum of free and complexed sulfate based on our model results (circles), and as only free sulfate (stars). Each measurement has been performed on a pool of a hundred to several hundreds of specimens. Values are compared to other culture experiments of foraminifera targeting specifically the CAS content of the tests (Paris et al., 2014: van Dijk et al., 2017; 2019), and, when available, $S/CO_3^{2-}$ as well. See Appendix B, Table B2 for details.

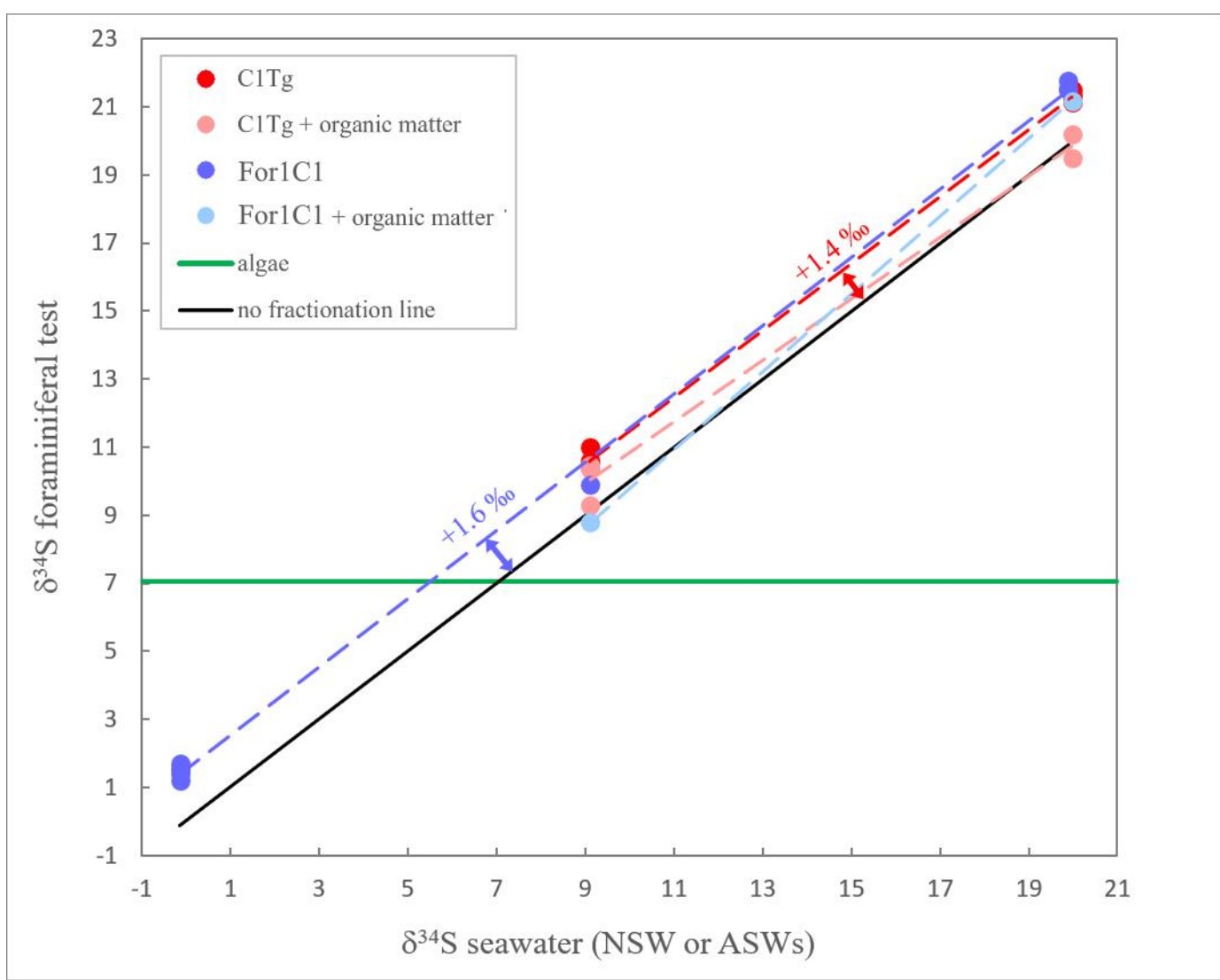

**Figure 8. δ³⁴S in foraminiferal test (with or without organic matter) for the two strains For1C1 and C1Tg. The two strains were cultured in several media (For1C1 in ASW[5], ASW[10] ASW[28], ASW[35], ASW[ 40], ASW[60] and NSW (Banyuls), C1Tg in NSW (Concarneau) and ASW[28]) whose δ³⁴S has been also measured. The δ³⁴S value of the media depend on the salts used to make the solution. The green line corresponds to the δ³⁴S composition of the algae that were fed to the foraminifera, and whose isotopic composition remain stable. 2σ error bars (±0.2‰ to ±0.3‰) are smaller than symbols.**


**Table 1. Weekly number of** accumulated live individuals incremented by reproduction **for each medium at different [SO$_4^{-2}$]**

| SET 1 | | Seawater [SO$_4^{2-}$] | | | | | | | |
|---|---|---|---|---|---|---|---|---|---|
| | | **0** | **5** | **10** | **NSW** | **28.2** | **60** | **120** | **180** |
| **Date** | **Days** | Number of specimens | | | | | | | |
| 23/03/2018 | 0 | 27 | 30 | 28 | 17 | 25 | 31 | 22 | 23 |
| 29/03/2018 | 6 | 27 | 60 | 61 | 17 | 63 | 36 | 22 | 23 |
| 04/04/2018 | 12 | 36 | 188 | 116 | 128 | 199 | 151 | 21 | 24 |
| 09/04/2018 | 17 | 21 | 227 | 187 | 294 | 317 | 98 | 13 | 17 |
| 18/04/2018 | 27 | 22 | 293 | 322 | 638 | 556 | 104 | 16 | 20 |
| 25/04/2018 | 34 | 20 | 425 | 713 | 732 | 1312 | 108 | 14 | 19 |

| SET 2 | | Seawater [SO$_4^{2-}$] | | | | | | |
|---|---|---|---|---|---|---|---|---|
| | | **1** | **10** | **28.2** | **35** | **40** | **50** | **90** |
| **Date** | **Days** | Number of specimens | | | | | | |
| 07/05/2018 | 0 | 6 | 6 | 6 | 6 | 6 | 6 | 6 |
| 14/05/2018 | 7 | 27 | 35 | 27 | 109 | 48 | 64 | 27 |
| 22/05/2018 | 15 | 82 | 109 | 138 | 141 | 97 | 89 | 32 |
| 28/05/2018 | 21 | 76 | 120 | 142 | 173 | 98 | 91 | 29 |
| 06/06/2018 | 31 | 87 | 737 | 151 | 444 | 127 | 97 | 25 |
| 08/06/2018 | 33 | 161 | 1014 | 159 | 470 | 194 | 117 | 17 |

**Table 2. pH values measured in media all along SET1 and SET2 experiments.**

**Set 1**

| Day | ASW[0] | ASW[5] | ASW[10] | ASW[28] | ASW[60] | ASW[120] | ASW[180] | Banyuls |
|-----|--------|--------|---------|---------|---------|----------|----------|---------|
| 0 | 8.09 | 8.09 | 8.08 | 8.1 | 8.1 | 8.14 | 8.14 | nd |
| 6 | 7.94 | 7.94 | 7.96 | 8.06 | 7.97 | 7.94 | 8.02 | nd |
| 7 | 8.17 | 8.19 | 8.19 | 8.19 | 8.19 | 8.17 | 8.2 | nd |
| 13 | 7.89 | 7.89 | 7.89 | 8.01 | 7.93 | 7.83 | 8.02 | 8.06* |
| 14 | 8.17 | 8.19 | 8.19 | 8.19 | 8.19 | 8.17 | 8.2 | nd |
| 25 | 8.1 | 8.1 | 8.09 | 8.13 | 8.15 | 8.16 | 8.2 | 8.07* |
| 35 | 8.05 | 8.07 | 8.07 | 8.11 | 8.12 | 8.14 | 8.16 | 8.03* |

**Set 2**

| Day | ASW[1] | ASW[10] | ASW[28] | ASW[35] | ASW[40] | ASW[50] | ASW[90] |
|-----|--------|---------|---------|---------|---------|---------|---------|
| 7 | 8.1 | 8.12 | 8.14 | 8.09 | 8.12 | 8.12 | 8.17 |
| 16 | 8.08 | 8.12 | 8.15 | 8.15 | 8.14 | 8.13 | 8.14 |

*: Foraminiferal culture in seawater from Banyuls started with a delay, making the pH measurements day 12, 22 and 26

**Table 3. DIC concentration in culture media all along Set 1 and Set 2 experiments**

| Days | 6 | 12 | 13 | 14 | 22 | 25 | 29 | 35 | 38 | 46 | 52 | 53 |
|---|---|---|---|---|---|---|---|---|---|---|---|---|
| Media | DIC mM +/- 4% | | | | | | | | | | | |
| ASW[1] | | | | | | | | | 3.5 | | | |
| ASW[5] | | | | 4.1 | | | | | | | | |
| ASW[10] | 3.5 | | 3.8 | | | 3.2 | | 3.6 | | | | 3.7 |
| ASW[28] | 3.9 | | 3.9 | | | 4.1 | | 3.8 | | | | 3.7 |
| NSW | | 3.5 | | | 3.7 | | 3.8 | | | | | |
| ASW[35] | | | | | | | | | 3.6 | 3.8 | 3.7 | |
| ASW[40] | | | | | | | | | 3.6 | | | |
| ASW[50] | | | | | | | | | 3.5 | | | |
| ASW[60] | | | | | | 4.1 | | | | | | |
| ASW[90] | | | | | | | | | | | 3.9 | |


**Table 4. Sulfate concentration and isotopic composition measured in the foraminiferal calcite**

| Species | Media | SO$_4$/CaCO$_3$ ppm +/- 5% | $\Delta^{34}$S* | $\delta^{34}$S CAS ±0.2‰ | $\delta^{34}$S SO$_4^{2-}$ water ±0.2‰ | [SO$_4^{2-}$] media (mM) |
|---|---|---|---|---|---|---|
| For1C1 | ASW[5] | 3320 | 1.7‰ | 1.6‰ | -0.1‰ | 5 |
| For1C1 | ASW[10] | 5780 | 1.5‰ | 1.4‰ | -0.1‰ | 10 |
| For1C1 | ASW[28] | 10800 | Nd | Nd | Nd | 28.2 |
| For1C1 | ASW[28] | 12800 | 0.8‰ | 9.9‰ | 9.1‰ | 28.2 |
| For1C1 | NSW | 14200 | 1.6‰ | 21.5‰ | 19.9 | 28.2 |
| For1C1 | ASW[35] | 13700 | 1.6‰ | 1.5‰ | -0.1 | 35 |
| For1C1 | ASW[40] | 14000 | 1.3‰[a] | 1.2‰ [a] | -0.1 | 40 |
| For1C1 | ASW[60] | 12800 | 1.8‰[b] | 1.7‰[b] | -0.1 | 60 |
| For1C1 + org | NSW | 20100 | 1.2‰ | 21.2 | 20.0 | 28.2 |
| For1C1 + org | ASW[28] | 9700 | -0.3‰ | 8.8 | 9.1 | 28.2 |
| C1Tg | NSW | 15600 | 1.3‰ | 21.3 | 20.0 | 28.2 |
| C1Tg | NSW | 13400 | 1.2‰ | 21.1 | 20.0 | 28.2 |
| C1Tg | ASW[28] | 10700 | 1.5‰ | 10.6 | 9.1 | 28.2 |
| C1Tg | ASW[28] | 10400 | 1.3‰ | 10.4 | 9.1 | 28.2 |
| C1Tg | ASW[28] | nd | 1.9‰ | 11.0 | 9.1 | 28.2 |
| C1Tg + org | NSW | 25600 | -0.8‰ | 192 | 20.0 | 28.2 |
| C1Tg + org | NSW | 32900 | 0.3‰ | 20.3 | 20.0 | 28.2 |
| C1Tg + org | ASW[28] | 12400 | 0.2‰ | 9.3 | 9.1 | 28.2 |

*$\Delta^{34}$S = $\delta^{34}$S CAS - $\delta^{34}$S SO$_4^{2-}$ water

[a] The 2sd value of this sample is estimated to be 0.25 ‰.

[b] The 2sd value of this sample is estimated to be 0.35 ‰.


# 6 Appendices

## 6.1 Appendix A

**Foraminifer taxonomy**

The two selected strains come from two distinct locations, from Banyuls (Mediterranean Sea) and Concarneau (Atlantic Ocean). Morphologically they may be related to the family Rosalinidae (Holzmann and Pawlowski, 2017). They are attached forms with a low trochospiral hyaline calcitic perforate test, with a peripherical low arch aperture on the umbilical side bordered by lips (Fig. 2). Chamber interior is simple (Fig. 2). Two morphotypes are noticeable in both the strains (Fig. 2). Individuals usually reproducing asexually after every 12-15 days when their test reaches a development of 11-12 chambers (Fig. 3), are morphologically very close to the genus *Rosalina* (Fig. 2). Individuals who lived for several weeks adding more than 12 chambers have the last chambers with an annular arrangement (Fig. 2).

Individuals with a *Rosalina* like morphology (Fig. 2) probably belong to the schizont generation of their trimorphic life cycle (alternating gamont-agamont-schizont-gamont generations), documented for example in *Planorbulina mediterranenis* and a few dozen other species (Le Calvez, 1938; Dettmering et al., 1998). More precisely, they are diploid megalospheric schizonts that have entered a cycle of successive asexual reproduction (apogamic cycle) (Fig. 3), during which the new generation of schizonts is produced by schizogony, i.e. by multiple fission of a multinucleate parental cytoplasm (Le Calvez, 1938; Dettmering et al., 1998). For this reason, it is not obvious to identify them morphologically at the species level because the morphology of the diploid agamont microspheric and/or of the haploid megalospheric gamont parent generation, on which the description of the species has often been made, is unknown to the best of our knowledge. For now, we leave these forms in open nomenclature and call them by the name of the strains For1C1 and C1Tg. Adult specimens of these strains are smaller than the traditional foraminifer fraction obtained after sieving (through >125 µm mesh) in geochemical studies, they thus may be common, while rarely collected because of their size.

## 6.2 Appendix B – supplementary tables

**Table B1. Sulfur isotope composition of media and algae cells**

| Sample | $\delta^{34}S$ +/- 0.2 ‰ |
| --- | --- |
| NSW Banyuls before culture | 21.1 |
| NSW after feeding For1C1 | 19.9 |
| ASW[28] before culture | 9.1 |
| ASW[28] after feeding For1C1 | 9.1 |
| ASW[28] after feeding C1Tg | 9.2 |
| ASW(all used concentrations except 28) after feeding For1C1* | -0.1 |
| Algae media | 5.4 |
| Algae cells | 7.0 |

* Different salts were used to make all ASW and ASW28


**Table B2. Comparison between different culture experiments. Our S/CO$_3^{2-}$ ratios are given at 15% RSD based on the replicates of the model results at 28 mM and 10 mM. S$^*$ represents free uncomplexed suflate in the solution.**

| Species | | S/Ca mmol/mol | SO$_4^{2-}$/CaCO$_3$ ppm | Tested parameter | [SO$_4^{2-}$] mM r | S/CO$_3^{2-}$ mol/mmol | S$^*$/CO$_3^{2-}$ | Ref. |
|---|---|---|---|---|---|---|---|---|
| *A. lessonii* | min | 1.21 | 1161 | T°C | 28 | n.a. | n.a. | van Dijk et al., 2019 |
| | max | 1.73 | 1660 | | 28 | | | |
| *S. marginalis* | | 8.95 | 8588 | pCO$_2$ | 28 | 0.12 | n.a. | van Dijk et al., 2017 |
| | | 9.6 | | | 28 | 0.14 | n.a. | |
| | | 10.4 | 9979 | | 28 | 0.18 | n.a. | |
| *A. gibbosa* | | 0.95 | 912 | pCO$_2$ | 28 | 0.12 | n.a. | van Dijk et al., 2017 |
| | | 1.02 | 979 | | 28 | 0.14 | n.a. | |
| | | 1.1 | 1056 | | 28 | 0.18 | n.a. | |
| | | 1.3 | 1247 | | 28 | 0.29 | n.a. | |
| *A. lessonii* | min | *1.27* | 1219 | salinity | 28 | 0.12 | n.a. | van Dijk et al., 2017 |
| | max | *1.35* | 1295 | | 28 | 0.13 | n.a. | |
| *O. universa* | | 1.11 | 1062 | [SO$_4^{2-}$] | 18 | n.a. | n.a. | Paris et al., 2014 |
| | | 1.21 | 1164 | | 21 | n.a. | n.a. | |
| | | 1.34 | 1289 | | 24 | n.a. | n.a. | |
| | | 1.72 | 1651 | | 28 | n.a. | n.a. | |
| For1C1 | | 3.46 | 3320 | [SO$_4^{2-}$] | 5 | 0.12 | 0.04 | This study |
| | | 6.02 | 5781 | | 10 | 0.44 | 0.15 | |
| | | 14.80 | 14200 | | 28 | n.a. | n.a. | |
| | | 11.25 | 10800 | | 28 | 0.81 | 0.35 | |
| | | 14.22 | 13651 | | 35 | 1.03 | 0.38 | |
| | | 14.59 | 14004 | | 40 | 1.2 | 0.45 | |
| | | 13.38 | 12841 | | 60 | 1.51 | 0.59 | |

### 6.3 Appendix C


**Potential experimental bias**

We designed Set 2 to replicate the ASW[28] and ASW[10] results as well as to extend the range of concentrations and ran it right after Set 1 (see methods for more details). Two differences can be observed. First the reproduction rate is significantly higher in the ASW[10] media of Set 2 than it was in Set 1 even though the media were identical (Fig. 5). However, it might be related to the starting number of foraminifera for each experiment, in Set 2 we started each culture experiment with 6 foraminifera individuals (instead of 28 as in Set 1), each 6 individuals were chosen more carefully, which could induce a bias and explain a more active behavior during experiment 2. If it were the case, the bias would nonetheless be systematic and similar for each medium (all Petri dishes in set 2 started with 6 individuals) and thus do not prevent comparison of results within set 2.

The second difference is observed in the ASW[28] medium in set 2. The reproduction rate, which was the highest observed, slowed down drastically after 15 days. This decrease can be explained by a microbial bloom in the media that was observed in no other media (Fig. C1). The microbial spread could not be reduced by the weekly water change, and any transfer and rinsing of

foraminifera or antibiotic treatment would have constituted an additional experimental modification. We thus kept counting foraminifera and sampling seawater, but did not take into account any results collected in that media after day 15.


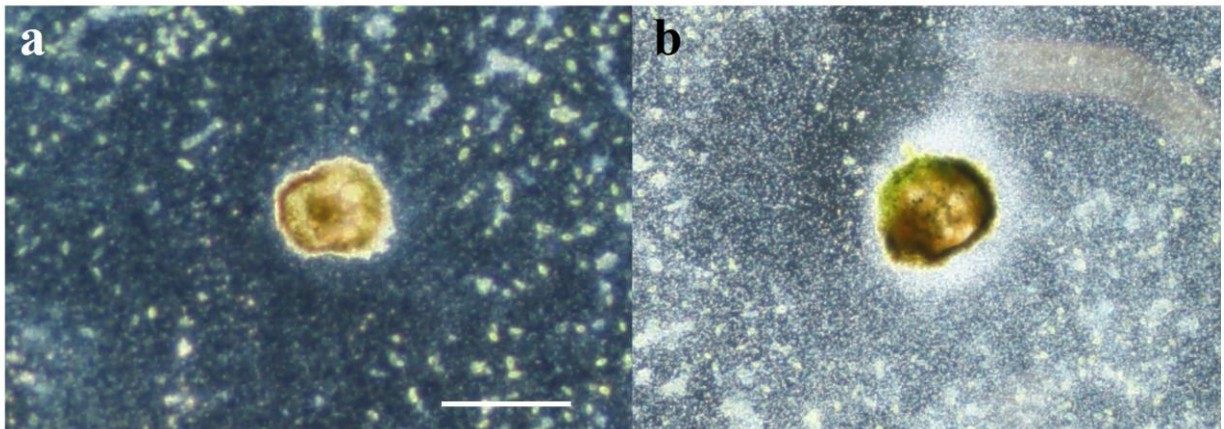

**Figure C1. Optical microscopy imaging in dark field, the foraminifera are observed from below, the background appears black, algae greenish, and bacterial contamination cloudy white. (a) Foraminifera in ASW[10] during experiment 2 where microbial spread stays**

**limited. (b) Foraminifera in ASW[28] during experiment 2 where microbial bloom was uncontrolled after 15 days and could not be reduced. Scale bar 100µm.**

## 6.3 Appendix D

### Geochemical modeling Parameters

The relative abundances of $Ca^{2+}$, $SO_4^{2-}$, $CaSO_{4(d)}$, $MgSO_{4(d)}$, $NaSO_4^{-}{}_{(d)}$ $CaHCO_3^+$, $CO_3^{2-}$, $HCO_3^-$ and $CO_{2(d)}$ in solution were computed with the geochemical code JCHESS (Van der Lee, 1998). ASW composition from kester et al., 1967 and experimental temperature (20 C) were chosen as input parameters assuming a closed system with no gas–solution exchange, and Cl, Na and $SO_4^{2-}$ were modified as they were in each media of the experiment (Base input in the additional excel table). All DIC is provided as $HCO_3^-$ to the model.

Given that pH was adjusted by NaOH addition, this was reproduced in the model, by adding $Na^+$ and $OH^-$ (in the same concentration) until reaching pH 8.2.
From this starting point, $HCO_3^-$ was adjusted to the measured DIC value, and $OH^-$ was adjusted to match the measured pH value (Adjusted input, and measured values in the additional excel table). No additional $Na^+$ was added despite the slight electrical imbalance generated, as $Na^+$ can form complexes with $SO_4^{2-}$ and no $Na^+$ was provided to the media after pH has been adjusted.

The output data considered are $Ca^{2+}$, all free and complexed $SO_4^{2-}$ species, free DIC species and $CaHCO_3^-$. The sole DIC specie present as a complex that was extracted is $CaHCO_3^-$, because it is the major complex. It is also a species that has been hypothesised to be potentially incorporated into calcite, as $CaSO_4$ could be.

### Geochemical modelling results

The $SO_4^{2-}/CO_3^{2-}$ concentration increases linearly to a slight inflection point at 60mM, linked to complexes formation. Nevertheless, we do not observe a plateau from 28mM onwards, which could have explained a constant incorporation of $SO_4^{2-}$ in calcite beyond 28mM. Similarly, assuming that $SO_4^{2-}$ incorporation into calcite takes place from $CaSO_4$, although an inflection of the $CaSO_4/CaHCO_3$ ratio is observable from 40mM, no plateau is observed. These results show that the incorporation of sulfate into the calcite of the foraminifers in our experiments, which plateau above 28mM, cannot be explained by the formation of com-

plexes in seawater. Alternatively, CAS concentration is not a good recorder of either the $SO_4^{2-}/CO_3^{2-}$ ratio or the $CaSO_4/CaHCO_3$ ratio above a concentration of 28mM.

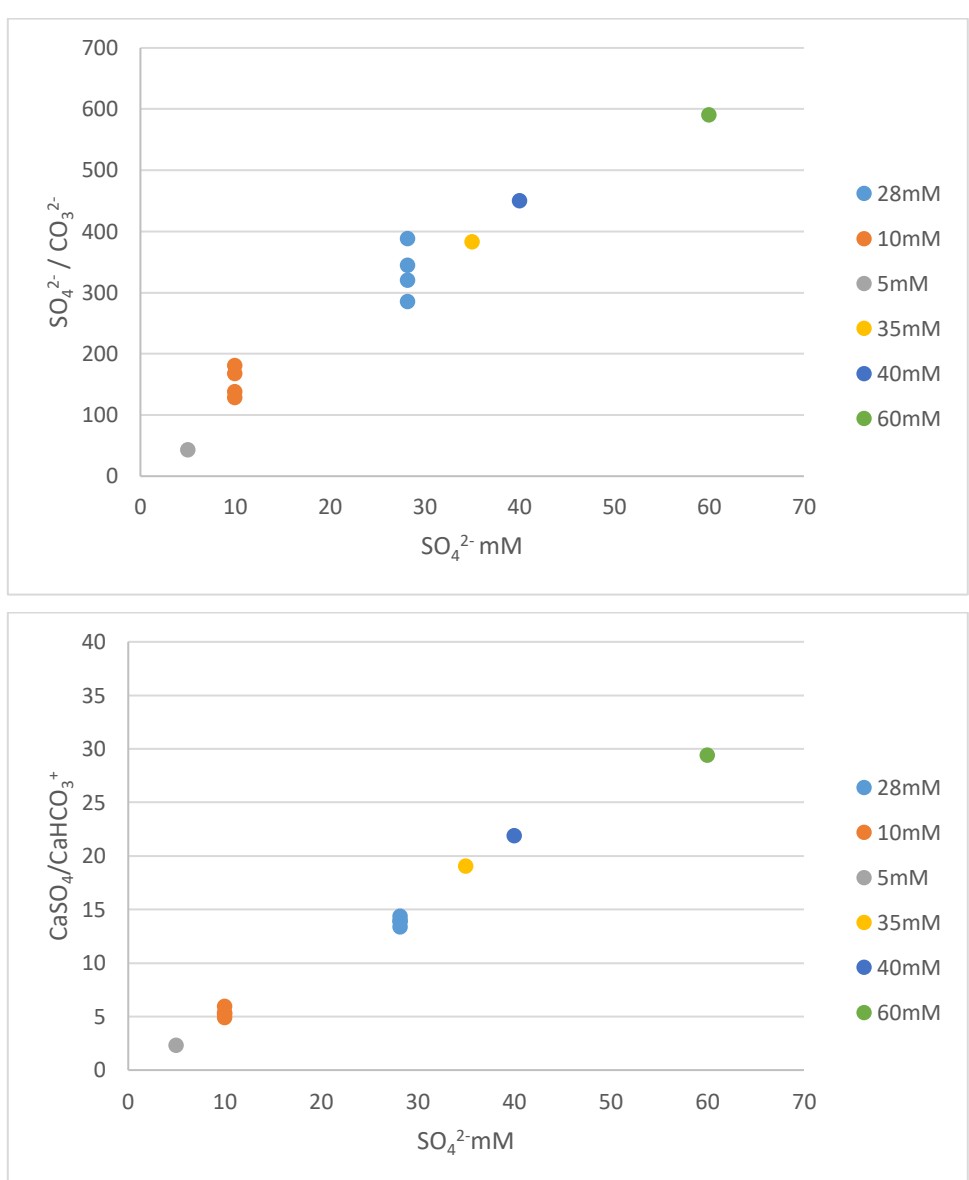

Figure D1. Geochemical modeling results showing $SO_4^{2-}/CO_3^{2-}$ ratio and $CaSO_4/CaHCO_3$ ratio in solution as a function of total sulfate concentration in solution (dots color indicate each sulfate concentration). Each point corresponds to a different computational run, for samples where DIC and pH where measured, and were used as constraints to the model. Both ratios increase linearly to a slight inflection point at 60mM, but no plateau is seen between 28mM and 60mM.