# Peer review of "Impact of seawater sulfate concentration on sulfur concentration and isotopic composition in calcite of two cultured benthic foraminifera"

_EGUsphere, 2023_

## Referee Comment (RC2)

The manuscript 'Impact of seawater sulfate concentration on sulfur concentration and isotopic composition in calcite of two cultured benthic foraminifera' by Thaler *et al*. presents laboratory culture experiments of a strain of Rosalinid foraminifera grown under different seawater sulphate concentrations and sulphur isotopic compositions. The rationale is that foraminiferal carbonate associated sulphate (CAS), i.e., sulphate that is presumably incorporated as an impurity into the mineral lattice, may be an archive of past changes in the seawater sulphate concentration and isotopic composition. To this end, the authors have performed a set of carefully-conducted and well-described experiments covering a wide range in $[SO_4^{2-}]$ and $\delta^{34}S$. The results show that, up to a point, the foraminifer CAS concentration is proportional to that in seawater, and that $\delta^{34}S_{shell}$ is related to $\delta^{34}S_{seawater}$ with a slope of 1 but an offset of 1.4-1.6 ‰. Together, these results provide promising indication that foraminiferal CAS is likely to be a useful archive of past changes in the sulphur cycle, while highlighting the need for calibration datasets to identify species-specific vital effects.

I have a few main comments, related to the possible impact of seawater carbonate chemistry on some of the results and whether or not the data presented here really enable us to say something about the organisms' ability to regulate sulphate within the cell or at the biomineralisation site via active transport. While I suggest rephrasing parts of the discussion with this in mind, the manuscript is interesting and presents an important dataset, which I look forward to seeing published.

**Main comments**
1. As the authors acknowledge, previous work has identified the seawater $SO_4^{2-}/CO_3^{2-}$ ratio as being a likely key control on S/Ca, as $SO_4^{2-}$ probably competes for the anion position in calcite. Given this, experimental work aiming to understand S incorporation must have excellent carbonate chemistry control, which is largely the case here (pH and DIC were measured and were broadly held constant). It does not appear that changes/variations in seawater carbonate chemistry within and between experiments is a major issue for this study, but I would suggest that the authors calculate $SO_4^{2-}/CO_3^{2-}$ for each experiment, and additionally (or alternatively) plot the results (Fig. 6) against this parameter. For example, the DIC in experiment ASW[60] was ~20% higher than in ASW[40]/ASW[50], which may explain the lower S/Ca ratio of this experiment.
2. In several places the manuscript contains inferences about biomineralisation which are very interesting but for which there is arguably little evidence based on the data presented here. For example, in Sec. 4.2, the possibility that foraminifera can regulate the biomineralisation site $[SO_4^{2-}]$ or actively maintain a constant $SO_4^{2-}/CO_3^{2-}$ is mentioned. Both seem unlikely to me, although it is clearly stated that these are hypotheses, which is of course fine. However, these hypotheses appear in more certain terms elsewhere (e.g. the abstract 'highlighting the extent of control on the precipitation fluid chemistry…' (line 26) which should be removed or qualified, and lines 30, 44, 301 (even within the framework of hypothesising, I don't think 'probably indicates' is accurate), lines 370-375, 381). All of these sentences/sections should be rephrased more cautiously in my view.
3. Adaptation. The importance of acclimatisation and the benefit of working with benthic foraminifera is highlighted in a couple of places in the manuscript (e.g. line 68), but it was not quite clear to me whether only specimens that grew entirely under experimental conditions were selected for analysis, and to what extent reproduction can be considered adaptation. Was the original population/were empty shells following reproduction removed before geochemical characterisation? Please clarify. If not, then the point about acclimatisation to experimental conditions doesn't stand.
4. Complexation is mentioned in the discussion, and I agree that it would be helpful to calculate speciation in these experiments. Indeed, doing so (using phreeqc; see figure) highlights that i) the

experimental design means that $\Omega_{calcite}$ decreased with increasing [SO$_4^{2-}$], and ii) that at the extreme high end of the [SO$_4^{2-}$] studied here, the seawater was possibly saturated with respect to gypsum/celestite. Again, I doubt this impacts any of the main conclusions, but it would be good to include a more thorough discussion of the topic rather than talking in very general terms (lines 311-314). For example, perhaps the decreasing $\Omega_{calcite}$ contributed to the lower growth/survival rates of foraminifera grown at higher [SO$_4^{2-}$], given that $\Omega_{calcite}$ approximately halves between the lowest and highest [SO$_4^{2-}$] experiments.

Finally, given that some of the experimental seawaters used here are close to being oversaturated with respect to gypsum and celestite, was any inorganic precipitation observed in the cultures? Gypsum precipitation seems unlikely, but e.g. if [Sr$_{sw}$] was a little higher than the assumed/target value, then this may have been an issue in the higher [SO$_4^{2-}$] experiments.

[Figure]

[Phreeqc calculations using the solution compositions given in Sec. 2.1.2, pH = 8.2, DIC = 4 mM, and the pitzer database. The Matlab code used to produce the figure is given on page 4.]

5. There isn't a lot of foraminifera S/Ca data out there, so it would be a nice addition to the manuscript (and not too much work) to compare directly to previous studies, especially that of van Dijk et al. [2017].

**Minor comments**
1. Line 23. Consider clarifying in the abstract why this range is different than on line 21 (the reason is given later, but to avoid confusion).
2. Line 36. Clarify that this is the case at higher (room) temperature.
3. Line 124. I didn't understand – it is stated that the second set was 'designed to extend our concentration range', but the range is narrower than in the first set.
4. Line 149. Table 3 alternatively states ±4%.
5. Section 2.2. Were foraminifera from set 1 and set 2 combined for analysis, where they were grown under the same conditions?

6. Section 2.3. I think it's fine to do so briefly, but please give some basic details of how the instruments were set up, exactly how the analyses were run, how the data were processed, what the blank data looked like etc. etc., rather than simply referring to a previous publication.
7. Line 222. I'm not sure that this range (= ~50% of the modern ocean [DIC]) could really be considered fairly stable. I was also missing an explanation of why DIC was much higher in these experiments than natural seawater.
8. Section 3.3/figures. It might be helpful to report molar S/Ca ratios, to maintain consistency with the vast majority of the geochemical literature and previous work on foraminifera S.
9. Line 248. Please state what the uncertainty represents.
10. Lines 274-275. On the other hand, over the range that $[SO_4^{2-}]$ is thought to have varied within the Phanerozoic (~5-30 mM) there is no relationship between reproduction (growth?) and $[SO_4^{2-}]$.
11. Lines 286-288. Were the foraminifera crushed prior to cleaning? If not (possibly even if so) then inter/intra-crystalline organics likely remained.
12. Lines 294-296, lines 326-327. I found the comparison a little simplistic as of course pH/DIC/$[Mg^{2+}]$/organics are also very important.
13. Line 297, lines 322-324. I would also add a possible kinetic effect to the list. If crystal growth rates are lower at higher $[SO_4^{2-}]$, as the inorganic work indicates, all else being equal, then a nonlinear seawater-shell relationship might be expected.
14. Line 315. Please rephrase. Kadan *et al.* studied coccolithophores, which have a completely different biomineralisation pathway (e.g. centred on transmembrane ion transport rather than seawater vacuolisation).
15. Line 381. Does it become lethal? Or does it simply prevent calcification?
16. Figure 6. Seawater $SO_4^2/Ca^{2+}$ is unitless, and the $/CaCO_3$ is unnecessary on the y axis.

**Typos**
1. Line 19. Calcifiers.
2. Line 321. van Dijk.
3. Line 578. de Nooijer.

```matlab
clear variables

% requires a phreeqc installation
iphreeqc = actxserver('IPhreeqcCOM.Object');    % create PHREEQC COM object
% load desired database (pitzer for seawater)
dirP = uigetdir('C:\Program Files\USGS','Select directory containing PHREEQC databases');
iphreeqc.LoadDatabase([dirP '\pitzer.dat']);

% SO4, Na, and Cl co-vary in these experiments
SO4in = 0:0.1:180;
NaIn = 479:-(479-402)/(size(SO4in,2)-1):402;
ClIn = 612:-(612-175)/(size(SO4in,2)-1):175;

outputData = NaN(size(SO4in,2),6);
for i = 1:size(SO4in,2)
    IPCstringCell= {'SOLUTION 1',  ...
        ['-temp ', num2str(22)],  ...
        '-units mmol/L',  ...
        '-density 1.025',  ...
        ['-pH ', num2str(8.2)], ...
        ['Ca ', num2str(10.3)],  ...
        ['Mg ', num2str(53)],  ...
        ['B ', num2str(0.4)],  ...
        ['K ', num2str(10)],  ...
        ['Br ', num2str(0.8)],  ...
        ['S(6) ', num2str(SO4in(i))],  ...
        ['Na ', num2str(NaIn(i))], ...
        ['Cl ', num2str(ClIn(i))],  ...
        ['Si ', num2str(0)],  ...
        ['Sr ', num2str(0.1)],  ...
        ['P ', num2str(0)],  ...
        ['F ', num2str(0.1)],  ...
        ['C(4) ', num2str(4)],  ...
    'SELECTED_OUTPUT',  ...
        '-molalities  CO3-2 HCO3- CO2 MgHCO3+ NaHCO3 CaHCO3+ MgCO3 NaCO3- CaCO3', ...
        '-activities  CO3-2 HCO3- Ca+2 NaHCO3 NaCO3- SO4-2', ...
        '-SI calcite anhydrite gypsum celestite', ...
        'soln false',  ...
        'pH true',  ...
        'sim false',  ...
        'state false',  ...
        'time false',  ...
        'step false',  ...
        'pe false',  ...
        'distance false'};
    IPCstring = sprintf('%s\n', IPCstringCell{:});

    iphreeqc.RunString( IPCstring );
    OUTphreeqSTRING = iphreeqc.GetSelectedOutputArray;

    % retrieve the data
    loc = find(strcmp(OUTphreeqSTRING,'m_CO3-2(mol/kgw)'));
    outputData(i,1) = OUTphreeqSTRING{2,(loc+1)/2};
    loc = find(strcmp(OUTphreeqSTRING,'si_calcite'));
    outputData(i,2) = OUTphreeqSTRING{2,(loc+1)/2};
    loc = find(strcmp(OUTphreeqSTRING,'si_anhydrite'));
    outputData(i,3) = OUTphreeqSTRING{2,(loc+1)/2};
    loc = find(strcmp(OUTphreeqSTRING,'si_gypsum'));
    outputData(i,4) = OUTphreeqSTRING{2,(loc+1)/2};
    loc = find(strcmp(OUTphreeqSTRING,'la_SO4-2'));
    outputData(i,5) = OUTphreeqSTRING{2,(loc+1)/2};
```

```matlab
        loc = find(strcmp(OUTphreeqSTRING,'si_celestite'));
        outputData(i,6) = OUTphreeqSTRING{2,(loc+1)/2};

end

% plot Omega calcite, anhydrite, gypsum, celestite
close(figure(1))
figure(1)
plot(SO4in,10.^(outputData(:,2)))
hold on
plot(SO4in,10.^(outputData(:,3)))
plot(SO4in,10.^(outputData(:,4)))
plot(SO4in,10.^(outputData(:,6)))
set(gcf,'color','w')
xlabel('[SO_4^{2-}] (mM)')
ylabel('\Omega')
legend('calcite','anhydrite','gypsum','celestite',...
    'location','southeast','fontsize',8)
set(gca,'yscale','log')
ylim([1e-2 20])
```

---

## Author Comment (AC1)

**egusphere-2023-631_Review_JR_ Julien Richirt**

The study investigates the impact of various seawater sulphate concentrations on carbonate associated sulphate (CAS) content and sulphur isotopic composition of a benthic foraminifera morphotype. To do so, the authors cultured for ~30 days specimens of *Rosalina*-like foraminifera from 2 strains coming from Mediterranean and Atlantic French coast in different media where the $[SO_4^{-2}]$ varied between 0 and 180 mM. Acclimatised for a long period of time before the experiment, CAS content and sulphur isotopic composition of their shell was measured after exposure to various $[SO_4^{-2}]$ concentrations. During the experiment, the abundance of living individuals was recorded in the different experimental conditions to evaluate their tolerance to the different $[SO_4^{-2}]$ in their surrounding environment. The general aim of the study is to explore the possibility to use foraminifera shell composition as a proxy for paleoenvironmental reconstructions of $[SO_4^{-2}]$ in seawater.

I found the manuscript interesting to read, and I think experimental approaches of this kind, while very difficult to realise in practice, are necessary to better understand how Foraminifera respond to their environment. I think this work can be very useful for the community regarding potential future $[SO_4^{-2}]$ proxy calibrations using foraminifer's test, and more generally concerning S-cycle understanding. This manuscript in the scope of the journal and is suited for publication in Biogeosciences.

However, I had some difficulties to comprehend certain parts of the manuscript and I sometimes did not understand fully interpretations and/or conclusions made by the authors. In some places I think that some sentences should be rephrased and that some terms are either misused or too vague. I list below major points that I consider important to be addressed by the authors before publication.

*We thank Dr. Richirt for this supportive comment. We have worked to reply to all of his questions and gave a lot of attention to rephrasing sentences that could have been considered as unsufficiently precise. We believe that the manuscript benefitted from his comments and suggestions and we hope that we have successfully addressed his concerns.*

1) The experimental protocol contains a lot of steps, and it took me a while before understanding it correctly. I think that the manuscript would highly benefit of either a workflow graph replacing table 1, or another possibility would be to indicate more details in table 1, such as: sampling year for each strain, number of years maintained in NSW, time of acclimation, number of specimens per condition, duration of each experiment, days of abundance counting… All these informations are available in the plain text and/or in other tables/graphs, but I think that to put everything in 1 graph (or table) would really make it easier for the reader to grasp all the efforts the authors put in the experimental settings.

*We have followed this suggestion and now a new Figure 1 showing the experimental protocol workflow replaces the old Table 1.*

[Figure]

2) On results & interpretations regarding foraminiferal abundances in experiments.

a. At the end of section 3.1, you compare abundances of individuals in the different media at the end of the experiment. Line 209-210, you state that the highest abundances were found for ASW28, NSW, ASW10 and ASW35. While the statement in absolutely correct, I would also consider in this list ASW5 (425 ind.) which is very close to the condition ASW35 (470 ind.).

*We have amended the text as suggested and added the ASW5 configuration to the list:" Overall, the highest numbers of individuals at the end of the experiment were obtained in the ASW[28], NSW (Banyuls), ASW[5], ASW[10] and ASW[35] media (Fig. 5 = old Fig. 4)."*

b. At the end of the same paragraph (lines 210-215), you try to explain the difference you observe between ASW10 in set1 and set2. I find the explanation unclear and probably unnecessary. For instance, what do you mean by "more carefully" about the individual selection before the experiment? Another reason why reproduction rate could be different between the 2 sets is that they just have more space in the Petri to multiply more efficiently, or less competition for food (if you put the same amount in all Petri)? Finally, abundances for ASW10 in both sets are of the same order of magnitude. Your graphical representation (log scale, fig. 4) is suitable to visualise this, the conditions 10mM from the 2 sets show relatively similar values regarding the sampling date. However, what must be discussed here (which is done in supplementary material appendix C) is the rather surprising low abundances of condition 28.2 set 2 compared with condition 28.2 set 1. I suggest mentioning it in section 3.1 instead of mentioning it only in appendix.

*In accordance with this suggestion, we have made the following changes to the text:"Two media configurations, ASW[10] and ASW[28], from set 1 experiments were replicated in set 2 experiments. If the abundances for condition ASW[10] are of the same order of magnitude in both sets, the abundances for condition ASW[28] are much lower in set 2 compared to set 1. This was related to the reproduction rate in set 2, which slowed down drastically after 15 days. This decrease can be explained by a microbial bloom in the media that was observed in no other media (Appendix, Fig. C1). The microbial spread could not be reduced by the weekly water change, and any transfer and rinsing of foraminifera or antibiotic treatment would have constituted an additional experimental modification. We thus kept counting foraminifera and sampling seawater, but did not take into account any results collected in that media after day 15."*

3) On results & interpretations regarding $SO_4^{2-}/CaCO_3$ ratio.

a. In section 3.3 about CAS concentration, on lines 226-227 you list the conditions for which enough tests could be collected for CAS analyses. However, I miss ASW50 (117 ind.) and ASW1 (161 ind.) both having more specimens than ASW60 (108). Why couldn't you perform analyses for these 2 conditions?

*We have rephrased the sentence by deleting "for each medium when enough tests could be collected for analyses" and adding " as the other samples were unfortunately lost during the manipulations or were below the detection limits."*

b. I might have misunderstood here, but in the same section on lines 229-232, you state that there is a threshold at 14000 ppm $SO_4^{-2}/CaCO_3$ for the 40mM condition. However, in table 4 I see a value of 11600 ppm for this condition. Figure 4 is consistent with table 4 values. Same problem for ASW5 condition for which it is 3320 ppm in your text (l. 229) but 2740 ppm in your table 4 and fig. 6. Please clarify or correct these values, since it is a crucial point for your further discussion.

*There was an error in the annotation of the values in Table and Figure 4 (now Figure 5). We have now homogenized the CAS values presented in the text, tables and figures. We have reworded the paragraph to make it clearer, in agreement with the reviewer, that there is a plateau from 28 mM to 60 mM and probably a threshold effect from 28 mM, given the error bars of the values.*

4) You use the term "physiology" in the abstract and in section 4.1 and I find the term not suitable to use in the context of this study. It is very confusing for me here because you find living individuals in all conditions at the end of your experiments. This indicates that individuals survive in all conditions, suggesting that they can sustain their physiological activity (even partially) in all conditions. However,

as your results point out, their capacity to reproduce seems to be dependent on seawater [$SO_4^{-2}$]. I suggest being more specific in the text and replace "physiology" with what is actually evaluated (reproduction rate/pseudopodial activity/survival…).

For instance, on line 265 (first occurrence in the text excluding the abstract):

*"Thus, our results suggest that foraminifera can sustain their physiological activity only within a certain range of [$SO_4^{2-}$] …."*

Could be replaced by:

*"Thus, our results suggest that foraminifera can reproduce only within a certain range of [$SO_4^{2-}$] …."*

*We have followed this suggestion and replaced the term "physiology" with reproduction, pseudopodial activity, etc.*
*However, we added in the discussions, paragraph 4.1 "In particular high seawater [$SO42-$] (> 35 Mm) inhibit the foraminiferal proliferation by an undetermined toxic effect on the cellular physiology".*
*The reduction in reproduction implies that cellular physiology has been affected, although we do not yet know the exact mechanisms and effects.*

I would like to inform the editor that I am not comfortable with the evaluation of the isotopic fractionation methods and result interpretations since it is out of my field of expertise. For this reason, I cannot fully assess the relevance of this part in the manuscript.
I think that some points, especially data and conclusion drawn from them, must be clarified/corrected before publication. Consequently, I recommend major revisions for this manuscript.
I require my name to be attached to this review, since it is not double anonymised.
Julien Richirt

**Detailed comments and suggestions (associated with annotated pdf file)**

**Abstract**
Line 27: please replace the term "physiology" by a more specific term.
*Done*

**Introduction**
Line 62: "interrogates".
*Done*

Line 65: "over a 180-fold range of seawater …". This is unclear, please specify the range, from 0 to 180.
*Done*

**Materials and Methods**
Line 85: Please specify if 90mm is the diameter or radius of the Petri dish you used.
Done

Line 97: please specify why it is more suited for geochemical studies.
*There was a shortcut in the logic of the sentence, we have rewritten this part as follows:*
*" Live algae can have a major impact on the seawater carbonate chemistry system by reproducing and consuming CO2 through photosynthesis. As freshwater algae, the Chlorogonium cells died immediately in seawater, without undergoing lysis. This prevents those not eaten by foraminifera from spreading and/or being metabolically active and thus they do not influence the seawater chemistry conditions within the Petri dishes. The use of live freshwater instead of seawater algae to feed foraminifera is therefore an innovative approach that is particularly suited to long term culture experiments for the calibration of foraminiferal geochemical proxies, where seawater chemical conditions must be kept under control".*

Line 100: what do you mean by "every other week"?

*"Every other week" means "every two weeks".*

Line 112: Is it true for train C1Tg which was not cultured in different sulphate concentrations?
*To clarify this question, we now add the following text: "The C1Tg strain was only used for [$SO_4^{2-}$] and $\delta^{34}S$ composition measurements of specimens from media in ASW or NSW at the current seawater average [$SO_4^{2-}$] of 28 mM, whereas the For1C1 strain was also used for [$SO_4^{2-}$] and $\delta^{34}S$ composition measurements of specimens from media with different [$SO_4^{2-}$]".*

Line 120: Table 1 lack some important informations that deserve to be in the main text and not in supplementary, such as number of individuals in each conditions.
*We have followed this suggestion and have now moved Table B1 from supplementary to the main text as new Table 1, and the sentence now refers to new Figure 1 (Experimental protocol workflow graph) and new Table 1 (Weekly number of accumulated live individuals for each medium at different $SO_4^{-2}$ concentrations).*

Line 127: What do you consider as a large population? When did you do this in the experiment course and for which conditions?
*We now specify in the text "For populations of more than approximately 300 individuals, as obtained in media with concentration ranging from 5 to 35mM sulfate"*

Line 128: "superpopulation" this term means something else, I guess you mean overpopulation?
*Done*

Line 128: "chlorogonium" should be in italic and capitalised.
*Done*

Line 129: "petri" sometimes capitalised sometimes not, please homogenise.
*We now homogenized all in capital*

Line 144-145: incomplete sentence
*We have now completed the sentence as follow: "The $CO_2$ and the He mix was then sampled with an autosampler and sent to a Dual Inlet FinniganTM DeltaPlus XP isotope ratio mass spectrometer"*

**Results**
Line 192: what do you mean by low cell density?
*We now specify: "where cells do not compete for food" and " less than approximately 300 individuals per Petri dishes" in our case*

Line 200: you could rephrase "number of individuals produced by the same cell" by "number of juveniles produced by individual". Done*

Lines 209-215: see my general comment 2).
*We have amended this part in line with comment 2.*

Line 219: in table 2 for ASW[5] I see ±0.3, not ±0.2
*We have now corrected and replaced 0.2 with 0.3*

Lines 226-234: see my general comment 3).
*We have now corrected the CAS values in Fig. 7 (old Fig. 6) and Table 4 according to the text.*
*Following JR's suggestion in the annotated pdf of the manuscript, we have now standardised that the threshold effect in foraminiferal CAS is reached at about 28mM seawater sulfate concentration.*

Lines 240: if you state that a difference is significant between 2 values, you should provide the statistical procedure you applied.

"NSW δ³⁴S composition was measured before (21.1±0.2‰) and 7 days after adding the algae (19.9±0.2‰). There was a significant difference between the two values."
*We have now added in the last sentence "beyond error bars"*

**Discussion**
Line 262: "dissolved sulfate in seawater is necessary for cellular activity in foraminifera". Out of curiosity, do you have any idea why they need sulfate from their surrounding water?

*Sulfur is essential for life, and sulfate in seawater, which is very abundant, can be an important source of sulfur for marine organisms. In the sentence: "In this experiment, dissolved sulfate and food were the only sources of sulfur", we added: "which is essential for life".*

Lines 265-266: see general comment 4). What I see on fig 4 is that, excluding the first reproduction event, condition 90 of set 2 is relatively similar to conditions 0, 120, 180 of set 2. Could it be that individuals might tolerate 90mM condition for 1 week and then stop any reproductive activity?
*In line with this comment, we have now added to the text "Individuals appear to tolerate these extreme conditions for only the first week and then cease all reproductive activity".*

Line 267: "In this experiment", I guess you mean the whole experiment, both sets? Please specify.
*We have now specified as both experiment sets*

Line 267: "weekly accumulated" what does that mean? aren't experiments running for about 1 month? Or is it the number of new living individuals added for each week? In this case what weeks are you looking at? Did you consider absolute abundances or proportion increase?
*For "weekly accumulated" individuals, we meant the number of new living individuals accumulating each week, incremented by reproduction, and we consider as absolute abundances (see table 2, ex table B2). In this part we mean at the end of each set of experiments. We have now better specified "the high number of accumulated the high number of accumulated live individuals incremented by reproduction at the end of set 1 and set 2 experiments"*

Line 268: "in both artificial and natural seawater" True only for set 1. If you consider condition 10mM of set2 (1014 ind. at 33 days) and NSW of set1 (732 ind. at 34 day), then this conclusion is incorrect. If you consider the number of individuals added week after week, then condition 10mM of set 1 also show a growth of about 300 ind. between 18/04 and 25/04, but I have the impression that I did not really get what you mean here.
*We have now corrected this part according this comment. As also presented in the results we have now wrote that individuals of the For1C1 strain appear to be well adaptable, beyond the modern oceanic [SO4²⁻] (28.2 mM) to a range of seawater [SO4²⁻] from 5 to 35 mM, as shown by the high number of accumulated live individuals at the end of set 1 and set 2 experiments*

Line 268: "suggesting that these species are highly adapted to their actual environment". I assume that by actual you mean modern. This suggestion is difficult to believe because the first part of the sentence is unclear.
*We have now changed this part in accordance with the previous comment*

Lines 269-270 and 274: please change the term physiology.
*We have changed the term physiology as suggested by the reviewer.*

Line 295: This is true for the first week, after that the number of individuals does not seem to grow much more. They might only tolerate short exposure to such high sulphate in their environment?
*Following this comment, we have now added "However, their reproduction is limited to the first week, which shows that they could only tolerate brief exposure to such a high level of sulfates in their environment."*

Line 305: please correct the author name in reference, also on lines 321 and 365.

*Done*

Lines 308: remove comma.
*Done*

Line 319: see general comment 3). Seeing fig 4, the moment they stop to incorporate more sulphate is about 30mM, not 40mM.
*The reviewer is probably referring to Figure 6 and not Figure 4. We have corrected the text to replace "40 mM" with "above 28 mM".*

Lines 342: I might be wrong here, but is not this organic sulphur source from algae? This value is 7 in fig. 7, table B2, and main text one line 241.
*It is actually related to the value of algae, we corrected and replaced it with the value 7.*

Lines 359-360: the CAS concentration reaches a plateau at about 30mM, not 40mM. see general comment 3).
*We have corrected this part, also taking into account the comments noted in the pdf, as follows: "Our results show that benthic foraminifera (Rosalinidae) incorporate CAS in their test proportionally to the $[SO_4^{2-}]$ in seawater, confirming previous experiments on planktic foraminifers that foraminiferal CAS can serve as a proxy for variations of both $\delta^{34}S_{CAS}$ and $[SO42-]$ in seawater (Paris et al., 2014). However, they also highlight that above the seawater $[SO_4^{2-}]$ of 28 mM, it is not possible to confidently determine the seawater $[SO_4^{2-}]$ using foraminiferal CAS, as the previous linear correlation no longer holds."*

Lines 368: "can affect the production of carbonate by affecting the biology of certain organisms". Too vague and too strong statement. I guess you mean reproduction rate/survival of your strain of benthic foraminifera? I recommend to at least change "can" for "could" to weaken this rather too strong statement!
*We corrected according the suggestion of reviewer and we wrote as follow : "could affect the production of carbonate by affecting the reproduction rate/survival of certain organisms, as in the case of the benthic foraminifera studied in this work."*

Lines 370-374: very difficult to understand this sentence, simplify or make 2 sentences.
Following this comment, we have simplified and made 2 sentences as follows: "This work illustrated *how variations in seawater composition can have a dual effect on biomineralizing organisms. Conditions that inhibit calcite formation such as increases in marine concentrations of Mg2+ or SO42-, could have chemical "abiotic" effects on carbonates formation but could also affect biological processes involved in biomineralization."*

**Conclusion**
Lines 379: "…foraminiferal biology…" please be more specific
*We corrected and replaced "foraminiferal biology" with "foraminiferal reproduction"*

Lines 381: "concentrations above 90mM becomes toxic and lethal". This is not what your data are suggesting! You have living individuals in all your conditions (fig 4, tables B1 and B2), and their abundance is most of the time very close to the number of individuals you initially put in the Petri dish! This rather suggest that they can survive these high sulphate concentration but cannot reproduce!
*We have corrected according this comment: "Sulfate from seawater is necessary for the cellular activity of foraminifera, but at concentrations equal and above 90 mM it becomes toxic to them, as evidenced by cellular inactivity and reproductive arrest"*

**Figures & Tables**

Fig 4. The legend for the line color of set1. Could you reorganise the display by sorting it by concentration such as on the right panel (set2).

*We have followed this suggestion and now in the Figure 5 (old figure 4) the line color of set 1 is reorganized as those of set 1.*

Table 1. see general comment 1).

*We have followed this suggestion and now a new Figure 1 showing the experimental protocol workflow replaces Table 1.*

Table 4. for the strain C1Tg, SO4/CaCO3 values, I am surprised by the rather large variability of these values for NSW and ASW28 compared to the other strain. Did you used these data somewhere in the manuscript or other figures?

*There is indeed a reproducible difference in the concentration of CAS in one species to another, which is yet another argument in favor of species-specific calibration, as stated line 410.*

*"The use of CAS concentration as a marine [SO42-] record is thus still promising, despite the limitation discussed above, but will require calibration on various types of carbonates and species" .*

*We thus do not compare CAS concentration variations from one specie to the other in this paper.*

Table B1. I used these 2 table quite a lot during the review, I think showing these data in the main text might be useful for the reader. This is only a suggestion.

*We followed this suggestion and we moved the table B1 in the main text and named now as table 2*

Table B2. Please add unit

*Now table B1, we added ‰*

Appendix C. see general comment 2). The part about the microbial growth in the ASW28 condition set 2 is important to explain your results in my opinion and I would mention this in the main text rather than in appendix.

*We followed this suggestion*

---

## Author Comment (AC2)

**egusphere-2023-631_Review 2_E_ David_Evans**

The manuscript 'Impact of seawater sulfate concentration on sulfur concentration and isotopic composition in calcite of two cultured benthic foraminifera' by Thaler *et al.* presents laboratory culture experiments of a strain of Rosalinid foraminifera grown under different seawater sulphate concentrations and sulphur isotopic compositions. The rationale is that foraminiferal carbonate associated sulphate (CAS), i.e., sulphate that is presumably incorporated as an impurity into the mineral lattice, may be an archive of past changes in the seawater sulphate concentration and isotopic composition. To this end, the authors have performed a set of carefully-conducted and well-described experiments covering a wide range in [SO42-] and δ34S. The results show that, up to a point, the foraminifer CAS concentration is proportional to that in seawater, and that δ34Sshell is related to δ34Sseawater with a slope of 1 but an offset of 1.4-1.6 ‰. Together, these results provide promising indication that foraminiferal CAS is likely to be a useful archive of past changes in the sulphur cycle, while highlighting the need for calibration datasets to identify species-specific vital effects.

I have a few main comments, related to the possible impact of seawater carbonate chemistry on some of the results and whether or not the data presented here really enable us to say something about the organisms' ability to regulate sulphate within the cell or at the biomineralisation site via active transport. While I suggest rephrasing parts of the discussion with this in mind, the manuscript is interesting and presents an important dataset, which I look forward to seeing published.

*We thank Dr. Evans for this encouraging comment and took into account all of his comments as we detail afterwards. We hope that we have addressed all his concerns and thank him again for his review that contributed to improve our manuscript.*

**Main comments**

1.As the authors acknowledge, previous work has identified the seawater $SO_4^{2-}/CO_3^{2-}$ ratio as being a likely key control on S/Ca, as $SO_4^{2-}$ probably competes for the anion position in calcite. Given this, experimental work aiming to understand S incorporation must have excellent carbonate chemistry control, which is largely the case here (pH and DIC were measured and were broadly held constant). It does not appear that changes/variations in seawater carbonate chemistry within and between experiments is a major issue for this study, but I would suggest that the authors calculate $SO_4^{2-}/CO_3^{2-}$ for each experiment, and additionally (or alternatively) plot the results (Fig. 6) against this parameter. For example, the DIC in experiment ASW[60] was ~20% higher than in ASW[40]/ASW[50], which may explain the lower S/Ca ratio of this experiment.

*To reply to this useful comment, we have performed a computational geochemical modelling using the software JCHESS, available in the Appendix, taking into account our media configurations where we had CAS data. This model shows that the CAS concentration follows linearly the seawater $SO_4^{2-}/CO_3^{2-}$ or the seawater sulfate concentration, though the linearity of the relationship is different when we consider only free sulfate instead of total sulfate (free and in complexes) in the solutions.*
*However, formation of complexes in solution do not appear to explain the disappearance of the linear relationship between CAS and sulfate concentration in solution at high sulfate concentrations (see new figure 8).*

2. In several places the manuscript contains inferences about biomineralisation which are very interesting but for which there is arguably little evidence based on the data presented here. For example, in Sec. 4.2, the possibility that foraminifera can regulate the biomineralisation site [SO42-] or actively maintain a constant SO42-/CO32- is mentioned. Both seem unlikely to me, although it is clearly stated that these are hypotheses, which is of course fine. However, these hypotheses appear in more certain terms elsewhere (e.g. the abstract 'highlighting the extent of control on the precipitation fluid chemistry…' (line 26) which should be removed or qualified, and lines 30, 44, 301 (even within the framework of hypothesising, I don't think 'probably indicates' is accurate), lines 370-375, 381). All of these sentences/sections should be rephrased more cautiously in my view.

*We understand the reviewer's concern. As a result, all the sentences about the implications on biomineralisation have been rephrased more cautiously. :*

*In the abstract we replaced :*

*" Foraminiferal CAS concentration increased proportionally with [SO₄²⁻] concentration from 5 mM up to a threshold value of 40 mM, highlighting the extent of control on the precipitation fluid chemistry that foraminifera exert on the carbonate precipitation loci."*

*By :*

*"Foraminiferal CAS concentration increased proportionally with [SO₄²⁻] concentration from 5 mM up to 28 mM, and then showed a plateau from 28 to 60 mM. The existence of a threshold at 28 mM is interpreted as the result of a control on the precipitation fluid chemistry that foraminifera exert on the carbonate precipitation loci. However, at high seawater sulfate concentrations (> 40 mM) the formation of sulfate complexes with other cations, may partially contribute to the non-linearity of the CAS concentration in foraminiferal tests at high increases in [SO₄²⁻]."*

*And in the discussion we now state :*

*"i) Foraminifera may be able to regulate [SO₄²⁻] at the site of calcification (SOC) during calcite precipitation through active transmembrane transport, removing excess sulfate and lowering it in the precipitating fluid, enabling calcite nucleation and precipitation, as sulfate in high concentration inhibits calcite precipitation and makes it more soluble (Busenberg and Plummer, 1985; Bots et al., 2011; Barkan et al., 2020). In fact, under our experimental conditions the amount of CAS incorporated in foraminiferal calcite correlates with seawater SO₄²⁻ concentration, from 5 up to a plateau that starts at 28 mM. The mere fact that calcite precipitates therefore suggests that sulfate is at least partially removed from the precipitating fluid, altering the local SO₄²⁻ concentration. The correlation suggests that this removal is partial and, to some extent, proportional to the concentration of SO₄²⁻ in seawater. "*

3. Adaptation. The importance of acclimatisation and the benefit of working with benthic foraminifera is highlighted in a couple of places in the manuscript (e.g. line 68), but it was not quite clear to me whether only specimens that grew entirely under experimental conditions were selected for analysis, and to what extent reproduction can be considered adaptation. Was the original population/were empty shells following reproduction removed before geochemical characterisation? Please clarify. If not, then the point about acclimatisation to experimental conditions doesn't stand.

*We clarified by adding the sentence: "Only live individuals (not empty shells that were discarded in previous water changes) that had fully grown under the experimental conditions were selected for analysis."*

*We added also in the paragraph 2.2 "Collection and rinsing procedure of the tests for geochemical analyses" that "all live individuals of the strain For1C1, as specified above, those still attached to the substrate" from each Petri dish were recovered for geochemical analyses.*

4. Complexation is mentioned in the discussion, and I agree that it would be helpful to calculate speciation in these experiments. Indeed, doing so (using phreeqc; see figure) highlights that i) the experimental design means that Ωcalcite decreased with increasing [SO42-], and ii) that at the extreme high end of the [SO42-] studied here, the seawater was possibly saturated with respect to gypsum/celestite. Again, I doubt this impacts any of the main conclusions, but it would be good to include a more thorough discussion of the topic rather than talking in very general terms (lines 311-314). For example, perhaps the decreasing Ωcalcite contributed to the lower

growth/survival rates of foraminifera grown at higher [SO42-], given that Ωcalcite approximately halves between the lowest and highest [SO42-] experiments.

Finally, given that some of the experimental seawaters used here are close to being oversaturated with respect to gypsum and celestite, was any inorganic precipitation observed in the cultures? Gypsum precipitation seems unlikely, but e.g. if [Srsw] was a little higher than the assumed/target value, then this may have been an issue in the higher [SO42-] experiments.

[Phreeqc calculations using the solution compositions given in Sec. 2.1.2, pH = 8.2, DIC = 4 mM, and the pitzer database. The Matlab code used to produce the figure is given on page 4.]

*We did not observe any precipitation of gypsum/celestite in the culture media at the extreme high [SO$_4^{2-}$], but the formation of complexes and its impact on CAS concentration is a very important point. We have now added a geochemical model in the appendix, which takes into account our media configurations where we had CAS data. (This new Appendix D is available at the end of this paragraph).*

*We have also added in the discussion that this model shows that the model CAS concentration follows linearly the seawater SO42-/CO32- and CaSO4/CaHCO3 concentrations, which in turn depend mainly on the [SO42-] in solution, with a dip between 40 and 60mM, likely related to the formation of complexes. At sulfate seawater concentrations > 40 mM sulfate might complex more easily with other cations (Ca2+, K+, Mg2+, Na+, Sr2+) and such complexes cannot be effectively incorporated into the calcite lattice structure.*

*Regarding the effect of Ω calcite on foraminifers, at low sulfate concentration, where Ω calcite increases, we observe less reproduction and thus less calcite formation. We hence believe that calcite formation in our system is rather controlled by foraminifers biological state (that seems related to solution sulfate concentration in such extreme variations) and to a much lesser extent to Ω calcite in solution.*

**6.3 Appendix D**

815 **Geochemical modeling Parameters**

The relative abundances of $Ca^{2+}$, $SO_4^{2-}$, $CaSO_{4(d)}$, $MgSO_{4(d)}$, $NaSO_{4(d)}$ $CaHCO_3$, $CO_3^{2-}$, $HCO_3^-$ and $CO_{2(d)}$ in solution were computed with the geochemical code JCHESS (Van der Lee, 1998). ASW composition from kester et al., 1967 and experimental temperature (20 C) were chosen as input parameters assuming a closed system with no gas–solution exchange, and Cl, Na and $SO_4^{2-}$ were modified as they were in each media of the experiment (Base input in the additional excel table). All DIC is provided
820 as $HCO_3^-$ to the model.
Given that pH was adjusted by NaOH addition, this was reproduced in the model, by adding $Na^+$ and $OH^-$ (in the same concentration) until reaching pH 8.2.
From this starting point, $HCO_3^-$ was adjusted to the measured DIC value, and $OH^-$ was adjusted to match the measured pH value (Adjusted input, and measured values in the additional excel table). No additional $Na^+$ was added despite the slight electrical
825 inbalance generated, as $Na^+$ can form complexes with $SO_4^{2-}$ and no $Na^+$ was provided to the media after pH has been adjusted.
The output data considered are $Ca^{2+}$, all free and complexed $SO_4^{2-}$ species, free DIC species and $CaHCO_3^-$. The sole DIC specie present as a complex that was extracted is $CaHCO_3^-$, because it is the major complex. It is also a species that has been hypothesised to be potentially incorporated into calcite, as $CaSO_4$ could be.

830 **Geochemical modelling results**

The $SO_4^{2-}/CO_3^{2-}$ concentration increases linearly to a slight inflection point at 60mM, linked to complexes formation. Nevertheless, we do not observe a plateau from 28mM onwards, which could have explained a constant incorporation of $SO_4^{2-}$ in calcite beyond 28mM. Similarly, assuming that $SO_4^{2-}$ incorporation into calcite takes place from $CaSO_4$, although an inflection of the $CaSO_4/CaHCO_3$ ratio is observable from 40mM, no plateau is observed. These results show that the incorporation of sulfate into
835 the calcite of the foraminifers in our experiments, which plateau above 28mM, cannot be explained by the formation of complexes in seawater. Alternatively, CAS concentration is not a good recorder of either the $SO_4^{2-}/CO_3^{2-}$ ratio or the $CaSO_4/CaHCO_3$ ratio above a concentration of 28mM.

[Figure]

840

Figure D1. Geochemical modeling results showing $SO_4^{2-}/CO_3^{2-}$ ratio and $CaSO_4/CaHCO_3$ ratio in solution as a function of total sulfate concentration in solution. Each point corresponds to a different computational run, for samples where DIC and pH where measured, and were used as constraints to the model. Both ratios increase linearly to a slight inflection point at 60mM, but no plateau is seen between 28mM and 60mM.

5. There isn't a lot of foraminifera S/Ca data out there, so it would be a nice addition to the manuscript (and not too much work) to compare directly to previous studies, especially that of van Dijk et al. [2017].
*We have followed this suggestion by adding data from previous studies to Figure 7 and new Table B2 in the Appendix. We have also included a comparison of our data with those published in the literature in the introduction to section 4.2 of the Discussion.*

[Figure]

Figure 7. **Left panel:** SO₄²⁻/CaCO₃ and S/Ca ratios on tests of the For1C1 strain at the end of SET1 and SET2 experiments, as a function
of seawater [SO₄²⁻] (5, 10, 28.2, 35, 40 and 60 mM). **Right panel: SO₄²⁻/CaCO₃ and S/Ca ratios on tests of the For1C1 strain at the end of
SET1 and SET2 experiments, as a function of seawater S/CO₃²⁻ . For our experimental results, we report the values using both S as the
sum of free and complexed sulfate based on our model results (circles), and as only free sulfate (stars). Each measurement has been
performed on a pool of hundred to several hundreds of specimens. Values are compared to other culture experiments of foraminifera
targeting specifically the CAS content of the tests (Paris et al., 2014: van Dijk et al., 2017; 2019), and, when available, S/CO₃²⁻ as well.**
See Appendix B, Table B2 for details.

725

730

**Minor comments**

1.Line 23. Consider clarifying in the abstract why this range is different than on line 21 (the reason is given later,
but to avoid confusion).

*We clarified that while the benthic foraminifera were cultured under controlled conditions with seawater [SO42-
] ranging from 0 mM to 180 mM (line 21), we measured CAS and d34S (line 23) on samples from "**a selection of
culture media**" where [SO42-] varied from 5 to 60 mM.*

*And line 266 it is now specified: " CAS concentration in foraminiferal calcite was performed for the media
ASW[5], ASW[10], ASW[28], ASW[35], ASW[40] and ASW[60], as the other samples were unfortunately lost
during the manipulations or were below the detection limits".*

2. Line 36. Clarify that this is the case at higher (room) temperature.
*We clarified adding in the sentence "at room temperature"*

3. Line 124. I didn't understand – it is stated that the second set was 'designed to extend our concentration range',
but the range is narrower than in the first set.
*We have corrected this sentence (see also the reply to Julien Richirt's comments) as follows "designed to refine
the concentration step between 0 and 90 mM".*

4. Line 149. Table 3 alternatively states ±4%.
*It was a mistake that has now been corrected to 5%*

5. Section 2.2. Were foraminifera from set 1 and set 2 combined for analysis, where they were grown under the
same conditions?
*We have now specified this in the manuscript by adding the sentence: "Individuals from set 1 and set 2, grown
under the same conditions (same medium [SO₄²⁻]), were not combined for analysis. They were measured
separately".*

6. Section 2.3. I think it's fine to do so briefly, but please give some basic details of how the instruments were set up, exactly how the analyses were run, how the data were processed, what the blank data looked like etc. etc., rather than simply referring to a previous publication.

*We have now added more details on geochemical analysis methods*

7. Line 222. I'm not sure that this range (= ~50% of the modern ocean [DIC]) could really be considered fairly stable. I was also missing an explanation of why DIC was much higher in these experiments than natural seawater.

*We removed the "fairly stable" mention and only refer to the range. We added the following explanation for the high DIC concentration:*

*"These concentrations are higher than the theoretical initial concentration of 2.8 mM using the recipe of Kester et al. 1967. While in Kester et al.'s recipe, the targeted 8.2 pH is achieved after 2h equilibration with the $CO_2$ in the atmosphere, we had to proceed to NaOH addition despite a 12h equilibration time. It is possible that higher $CO_2$ dissolution at the atmospheric pressure of the year we performed the experiments (407 ppm against 322 ppm in 1967), led to an increase in DIC. In addition, DIC probably built up in the Petri dishes each week as the foraminifera consumed the algae."*

8. Section 3.3/figures. It might be helpful to report molar S/Ca ratios, to maintain consistency with the vast majority of the geochemical literature and previous work on foraminifera S.

*We followed this suggestion and we reported in the figure 7 and in the appendix the values as molar S/Ca ratios*

9. Line 248 "A $\delta 34S_{CAS}$ - $\delta 34S_{sw}$ fractionation value of 1.6 ±0.3‰ was observed for For1C1 pool (9 samples in total coming from all [$SO_4^{2-}$] concentrations) while it was 1.4±0.2‰ for C1Tg specimens (9 samples in total coming from NSW or ASW[28]), which is indistinguishable within the error range (Fig. 7)". Please state what the uncertainty represents.

*Thank you for pointing this out. The uncertainty is the 1sd calculated from each individual fractionation. We specify it in the text.*

10. Lines 274-275. On the other hand, over the range that [$SO_4^{2-}$] is thought to have varied within the Phanerozoic (~5-30 mM) there is no relationship between reproduction (growth?) and [$SO_4^{2-}$].

*In the range of Phanerozoic [$SO_4^{2-}$] variations (~5-30 mM), the reproduction and growth of populations seems to be rather optimal. We added the sentence: "However, this appears to be for seawater [$SO_4^{2-}$] variations far below and above the range (~5-30 mM) thought to be involved in long-term secular variations in the Phanerozoic, suggesting an adaptation of foraminifera in this range of variations. Indeed, under conditions that mimic the Phanerozoic range of [$SO_4^{2-}$] variations, reproduction and population growth appear rather to be optimal."*

11. Lines 286-288. Were the foraminifera crushed prior to cleaning? If not (possibly even if so) then inter/intra-crystalline organics likely remained.

*Foraminifera were not crushed. We agree with the reviewer that organics likely remained, as stated in the text: "we assume that most of the measured [$SO_4^{2-}$] in the tests are linked to the CAS concentration, although a small contribution might be still associated with $S_{org}$ within the biomineralized calcite". We also performed analysis of foraminifera with organic carbon, which permit to tell that traces of organic carbon could have lowered the overall $\delta^{34}S$ value. But there is no reason to believe that different amount of inter/intra crystalline organics would have been preserved from one concentration to the other. This study main conclusion is thus focusing on the*

*maintained fractionation factor between foraminiferal $\delta^{34}S$ to water sulfate $\delta^{34}S$ at each concentration, rather than discussing the absolute isotopic fractionation value.*

12. Lines 294-296, lines 326-327 I found the comparison a little simplistic as of course pH/DIC/[Mg2+]/organics are also very important.

*We tried to develop the discussion here (in blue) and added a few references:*

*"The putative mechanisms from i to iii for sulfate regulation could have been adopted by foraminifera as evolutionary strategies to maintain carbonate precipitation despite potential variation in [$SO_4^{2-}$]. Indeed, at [$SO_4^{2-}$] greater than 8 mM abiotic calcite nucleation and precipitation is inhibited, and aragonite precipitates from saturated solutions (Kitano and Hood, 1962; Kitano et al., 1975; Bots et al., 2011). This inhibition is also true in the lack in magnesium (Barkan et al., 2020) and thus sulfate alone can affect calcite precipitation. Mechanisms such as increasing calcium concentration, pH and/or saturation state (e.g. Zeebe and Sanyal, 2002; Nehrke et al., 2013; Evans et al., 2018), as well as the presence of organics, could help overcome such high concentration of sulfate. However, when it comes to magnesium, active removal is also an option (Bentov and Erez, 2006)."*

13. Line 297, lines 322-324. I would also add a possible kinetic effect to the list. If crystal growth rates are lower at higher [SO42-], as the inorganic work indicates, all else being equal, then a nonlinear seawater-shell relationship might be expected.

*We have now added a discussion of "kinetic effect" to the list, as follows: "A kinetic effect could also explain the non-linearity of the CAS concentration in foraminiferal tests with corresponding increases in [SO42-] above 28 Mm, as inorganic calcite precipitation experiments suggest a reduction in crystal growth rates at higher [SO42-]. However, it is worth noting that a decrease in precipitation rate can also be associated to a lower CAS content in inorganic calcite (Barkan et al., 2020). As a result, one could imagine that the change in sulfate concentration reflects a change in precipitation rate induced by different sulfate concentration in seawater and/or in the biomineralizing fluid. However, as calcite is more soluble and precipitates less easily at high sulfate concentration, we would expect an effect opposite to what we observe in the 5-40 mM part of our results. There could nonetheless be a contribution of the rate effect to the plateau we observe"*

14. Line 315. Please rephrase. Kadan *et al.* studied coccolithophores, which have a completely different biomineralisation pathway (e.g. centred on transmembrane ion transport rather than seawater vacuolisation).

*We have removed this part and this reference, and this part remains more focused on the formation of complexes in the precipitation liquid.*

15. Line 381. Does it become lethal? Or does it simply prevent calcification?

*We deleted lethal according also the comments of Julien Richirt.*

16. Figure 6. Seawater SO42/Ca2+ is unitless, and the /CaCO3 is unnecessary on the y axis.

*Thank you. We corrected the figure, though we preferred to keep the /CaCO3 to avoid ambiguity.*

**Typos**
1.Line 19. Calcifiers.

*Corrected*

2. Line 321. van Dijk.

*Corrected*

3. Line 578. de Nooijer.

*Corrected*

---

## Referee Report (RR1)

The manuscript was greatly improved since the first submission, and I have read it again with pleasure. I appreciated the addition of useful figures and supplementary modelling approaches well used in the discussion section. I also appreciated that all my previous comments were considered carefully, and I was satisfied with the authors answers. However, because authors made important changes compared to the previous one, I have additional remarks and comments regarding this new version. Here are the 2 main problems I currently see with the manuscript that must be overcome before publication:

1. While I found the section 4.1 very interesting, I think some parts are unclear regarding what you mean exactly by tolerance, survival, cellular activity, pseudopodial activity, reproduction and calcification. From your data, forams seem to tolerate (survive) all concentrations, because you can find individuals alive (same amount than at the start) at the end of all your experimental conditions. However, reproduction (and calcification? I guess they cannot reproduce without being able to calcify?) is hampered or even prevented by the absence or high sulfate concentrations in your experiment. The pseudopodial activity assessment is also not clear: did you considered all individuals without any cytoplasm movement as dead even if they were attached to the substrate? In all cases, a foraminifera considered as living should have some cellular activity (even strongly reduced, such as during dormancy for instance).
On the question why they would develop and maintain such a tolerance vis-à-vis relatively high sulfate concentrations (almost 2 times higher than during the whole Phanerozoic, meaning they were never exposed to such high concentrations), could micro-habitat variability explain this mechanism? For instance, you mention volcanic activity or hydrothermal vents as causes of high sulfate content in SW. Could it be that forams developed such a "good" tolerance because they sometimes (temporarily) have to deal with higher sulfate concentration in their micro-habitat than 28.2 mM? This is very speculative but might be interesting to mention, especially for the potential consequences on paleo-proxies.

2. The data presentation and interpretation regarding sulfate/calcite ratios as a function of SW sulfate concentration in sections 3.3 and 4.2. This concerns the value at which the plateau begins to occur. You indicate several different values in the main text at different places. For example: from 28 to 60mM in abstract and results for instance lines 26 and 269. From 35 to 60mM in discussion line 345. 5-40 mM in discussion as referring as the part where there is a linear correlation between sulfate/calcite and sulfate concentration is SW on line 395. While these value changes are making it very unclear trough the manuscript for the reader, I think that your data indicate a relatively clear linear relationship between 5 and 35mM with a very good r² (in the following graph NSW is omitted). For this reason, I do not understand why you stick to 28.2 mM as the threshold value for the plateau starting point (which is the most encountered value in the current manuscript). Seeing the error bars on fig 7 I would recommend maybe considering the 5-35 mM range because there is no difference between 35 and 40mM (then the plateau starts at 35mM). This is rather important to be clear and consistent on this through the paper because it impacts the clarity for the reader and your further interpretation/discussion.

[Figure]

Minor and supplementary/more detailed comments are annotated **in the following pdf file**. I think the paper is very interesting and I am looking forward for it to be published in Biogeosciences. I couldn't choose between minor/major revision because I think the manuscript is in between at the moment. In my opinion, the paper suffers from important problems (consistency and clarity of some parts mainly), but these problems are relatively easy to solve. Because I still have troubles to understand the new data addition and sometimes their interpretation, I recommend (by default) major revisions for the current version of the manuscript.

I require my name to be attached to this review, since it is not double anonymised.

Julien Richirt

[revised manuscript text omitted]

---

## Referee Report (RR2)

This is my second review of the manuscript *Impacts of seawater sulfate concentration on sulfur concentration and isotopic composition in calcite of two cultured benthic foraminifera* by Thaler *et al.*, which presents S/Ca and $\delta^{34}S$ measurements of two strains of rosalinid foraminifera cultured under different seawater $[SO_4^{2-}]$.

The authors have, in my view, adequately addressed all of the comments from the first round of review, resulting in an improved version of what was already a very interesting and comprehensive piece of work. I have some further minor comments for the authors to consider but would suggest that they do so at their own discretion.

**Main comment**
1. Reproduction/growth in culture. If I understand what is written on lines 79 and sections 2.1.2/3.1, dead individuals (empty shells, or those not attached to the petri dishes) were removed, leaving only live foraminifera. Is it therefore possible that foraminifera that did not die but remained dormant could be present in the final analyses? In most cases this is of course not important, as there was a large increase in the number of individuals during the experiment. A possible exception to this is the 60 mM experiment, in which the number increased from ~30 to ~100. Given that the interpretation of the $[SO_4^{2-}]_{sw}$-S/Ca$_{shell}$ plateau hinges on this experiment, I suggest adding a note to explain whether or not this datapoint unambiguously does not contain pre-experiment shell material, or if it could represent a mixture of shell material grown under normal seawater and experimental conditions (which could then be an alternate explanation for the plateau if the foraminifera from reproduction in the experiment were smaller). On a similar note, Table 1 gives the numbers of foraminifera in each experiment through time, but how many were discarded during the experiment? Does this provide evidence for multiple generations? Even in the 60 mM $[SO_4^{2-}]$ experiment?

**Minor comments**
1. Lines 26-29. An inhibitory kinetic effect would seem a more likely possibility to me, but I appreciate not every hypothesis can be listed in the abstract.
2. Line 51. Sulphur and magnesium are not trace elements in seawater.
3. Lines 58 and 451. I suggest using a different phrase to 'large volcanic events' as this possibly implies single eruptions, whereas Laakso *et al.* discuss large igneous provinces emplaced over thousands of years.
4. Lines 75-76. You could clarify that most studies that included material grown before culture attempt to account for this in some way, e.g. using size-mass relationships or labels.
5. Lines 117-118. Are the units mM or mmol/kg?
6. Lines 211-212. Were all samples run at the same concentration as the seawater standards? If not, does this approach potentially result in a reproducibility that is too low?
7. Line 246. Please clarify which results you are referring to.
8. Lines 258-259. I think this explanation is unlikely (e.g. the DIC had increased by day 5 (Figure. 6) and there was no further increase.
9. Section 4.1 title. Bear in mind that there were large covariations in seawater $[Na^+]$ and $[Cl^-]$. Worth mentioning in the discussion?
10. Section 4.2. At some point (in the introduction?) it would be helpful to mention the Mg/Ca of these foraminifera if it is known. If they reduce the Mg/Ca of the biomineralisation site compared to seawater then the considerations regarding the effect of seawater $[SO_4^{2-}]$

on $CaCO_3$ nucleation and precipitation will likely not apply/be more complicated than implied in some places in the manuscript (e.g. lines 338, 407).

11. Line 338. On a similar note, I would suggest rephrasing this sentence. There may have been no precipitation in those experiments but it does not mean it is not possible, e.g. if higher degrees of oversaturation were to be achieved. Likewise, I would not read anything into the 'fact that calcite precipitates' (line 369).
12. Line 355. Please clarify, the same as what? *Heterostegina*?
13. Lines 416-418. Given that you include speciation modelling, can you say which ion pairs become relatively more abundant?
14. Line 720. It also diffuses between the experiment and atmosphere.

**Typos**
1. Line 65. Change 'interrogates' to 'suggests'.
2. Lines 137. Experiments.
3. Line 184. Aliquots, or delete 'of the'.
4. Lines 314-315, 390. Change four instances of 'Mm'.
5. Line 693. Forms.
6. Line 728. On a pool of a hundred to…'.
7. Line 816 and Fig. D1, it should be $CaHCO_3^+$ (also $NaSO_4^-$).

```matlab
clear variables

% requires a phreeqc installation
iphreeqc = actxserver('IPhreeqcCOM.Object');    % create PHREEQC COM object
% load desired database (pitzer for seawater)
dirP = uigetdir('C:\Program Files\USGS','Select directory containing PHREEQC databases');
iphreeqc.LoadDatabase([dirP '\pitzer.dat']);

% SO4, Na, and Cl co-vary in these experiments
SO4in = 0:0.1:180;
NaIn = 479:-(479-402)/(size(SO4in,2)-1):402;
ClIn = 612:-(612-175)/(size(SO4in,2)-1):175;

outputData = NaN(size(SO4in,2),6);
for i = 1:size(SO4in,2)
    IPCstringCell= {'SOLUTION 1',  ...
        ['-temp ', num2str(22)],  ...
        '-units mmol/L',  ...
        '-density 1.025',  ...
        ['-pH ', num2str(8.2)], ...
        ['Ca ', num2str(10.3)],  ...
        ['Mg ', num2str(53)],  ...
        ['B ', num2str(0.4)],  ...
        ['K ', num2str(10)],  ...
        ['Br ', num2str(0.8)],  ...
        ['S(6) ', num2str(SO4in(i))],  ...
        ['Na ', num2str(NaIn(i))], ...
        ['Cl ', num2str(ClIn(i))],  ...
        ['Si ', num2str(0)],  ...
        ['Sr ', num2str(0.1)],  ...
        ['P ', num2str(0)],  ...
        ['F ', num2str(0.1)],  ...
        ['C(4) ', num2str(4)],  ...
    'SELECTED_OUTPUT',  ...
        '-molalities  CO3-2 HCO3- CO2 MgHCO3+ NaHCO3 CaHCO3+ MgCO3 NaCO3- CaCO3', ...
        '-activities  CO3-2 HCO3- Ca+2 NaHCO3 NaCO3- SO4-2', ...
        '-SI calcite anhydrite gypsum celestite', ...
        'soln false',  ...
        'pH true',  ...
        'sim false',  ...
        'state false',  ...
        'time false',  ...
        'step false',  ...
        'pe false',  ...
        'distance false'};
    IPCstring = sprintf('%s\n', IPCstringCell{:});

    iphreeqc.RunString( IPCstring );
    OUTphreeqSTRING = iphreeqc.GetSelectedOutputArray;

    % retrieve the data
    loc = find(strcmp(OUTphreeqSTRING,'m_CO3-2(mol/kgw)'));
    outputData(i,1) = OUTphreeqSTRING{2,(loc+1)/2};
    loc = find(strcmp(OUTphreeqSTRING,'si_calcite'));
    outputData(i,2) = OUTphreeqSTRING{2,(loc+1)/2};
    loc = find(strcmp(OUTphreeqSTRING,'si_anhydrite'));
    outputData(i,3) = OUTphreeqSTRING{2,(loc+1)/2};
    loc = find(strcmp(OUTphreeqSTRING,'si_gypsum'));
    outputData(i,4) = OUTphreeqSTRING{2,(loc+1)/2};
    loc = find(strcmp(OUTphreeqSTRING,'la_SO4-2'));
    outputData(i,5) = OUTphreeqSTRING{2,(loc+1)/2};
```

```matlab
        loc = find(strcmp(OUTphreeqSTRING,'si_celestite'));
        outputData(i,6) = OUTphreeqSTRING{2,(loc+1)/2};

end

% plot Omega calcite, anhydrite, gypsum, celestite
close(figure(1))
figure(1)
plot(SO4in,10.^(outputData(:,2)))
hold on
plot(SO4in,10.^(outputData(:,3)))
plot(SO4in,10.^(outputData(:,4)))
plot(SO4in,10.^(outputData(:,6)))
set(gcf,'color','w')
xlabel('[SO_4^{2-}] (mM)')
ylabel('\Omega')
legend('calcite','anhydrite','gypsum','celestite',...
    'location','southeast','fontsize',8)
set(gca,'yscale','log')
ylim([1e-2 20])
```

---

## Author Response (AR2)

Answers to reviewers, second round of reviews :

Reviewer Julien Richirt :

The manuscript was greatly improved since the first submission, and I have read it again with pleasure. I appreciated the addition of useful figures and supplementary modelling approaches well used in the discussion section. I also appreciated that all my previous comments were considered carefully, and I was satisfied with the authors answers. However, because authors made important changes compared to the previous one, I have additional remarks and comments regarding this new version.

We'd like to thank Julien Richirt for his attention to our proofreading work, and we're pleased that he's satisfied with it.

Here are the 2 main problems I currently see with the manuscript that must be overcome before publication:

While I found the section 4.1 very interesting, I think some parts are unclear regarding what you mean exactly by tolerance, survival, cellular activity, pseudopodial activity, reproduction and calcification. From your data, forams seem to tolerate (survive) all concentrations, because you can find individuals alive (same amount than at the start) at the end of all your experimental conditions. However, reproduction (and calcification? I guess they cannot reproduce without being able to calcify?) is hampered or even prevented by the absence or high sulfate concentrations in your experiment. The pseudopodial activity assessment is also not clear: did you considered all individuals without any cytoplasm movement as dead even if they were attached to the substrate? In all cases, a foraminifera considered as living should have some cellular activity (even strongly reduced, such as during dormancy for instance).

We considered as dead all individuals that were detached from the substrate, but recognised that some rare foraminifera may still be attached after the death, as stated in line 143:

“We counted live individuals each week, for each medium. Since the studied species live attached to a substrate, individuals that no longer stick to the Petri dishes were considered dead, even though rare dead individuals (empty tests or no reticulopodial activity) may remain attached and few living adults can detach themselves from the substrate as well.”

On the question why they would develop and maintain such a tolerance vis-à-vis relatively high sulfate concentrations (almost 2 times higher than during the whole Phanerozoic, meaning they were never exposed to such high concentrations), could micro-habitat variability explain this mechanism? For instance, you mention volcanic activity or hydrothermal vents as causes of high sulfate content in SW. Could it be that forams developed such a “good” tolerance because they sometimes (temporarily) have to deal with higher sulfate concentration in their micro-habitat than 28.2 mM? This is very speculative but might be interesting to mention, especially for the potential consequences on paleo-proxies.

We doubt that our study permits to establish that this ability to tolerate variations in sulfate concentration was developed to deal with higher sulfate concentration. This is an interesting point, yet far beyond the scope of the current study. However, now that we know that this tolerance exists, we think it's important to take this feature into account when using foraminifera for paleoenvironmental reconstructions.

The consequences for paleoproxies is mentioned in the text:

"This limitation means that foraminiferal CAS could be used to trace deep time secular changes in seawater $[SO_4^{2-}]$, which varies from about 5 mM to 28 mM today (Algeo et al. 2015), but not to trace past seawater $[SO_4^{2-}]$ enrichments above 28 mM, such as those that could be caused by large volcanic eruptions or sulfate-rich volcanic hydrothermal fluids on the seafloor.

In addition, to be cautious, we have written that this threshold could be specific to Rosalinidae, see in 4.4 section of discussion: "Future works are therefore important to confirm whether or not the seawater [SO42-] threshold of 28 mM for CAS incorporation can be applied to other benthic and planktonic foraminifera, or whether it is restricted to Rosalinidae".

2. The data presentation and interpretation regarding sulfate/calcite ratios as a function of SW sulfate concentration in sections 3.3 and 4.2.

This concerns the value at which the plateau begins to occur. You indicate several different values in the main text at different places.

For example: from 28 to 60mM in abstract and results for instance lines 26 and 269.

From 35 to 60mM in discussion line 345.

5-40 mM in discussion as referring as the part where there is a linear correlation between sulfate/calcite and sulfate concentration is SW on line 395.

We are going to correct this, now it's stated everywhere: from 5 to 28 mM

While these value changes are making it very unclear trough the manuscript for the reader, I think that your data indicate a relatively clear linear relationship between 5 and 35mM with a very good r² (in the following graph NSW is omitted). For this reason, I do not understand why you stick to 28.2 mM as the threshold value for the plateau starting point (which is the most encountered value in the current manuscript). Seeing the error bars on fig 7 I would recommend maybe considering the 5-35 mM range because there is no difference between 35 and 40mM (then the plateau starts at 35mM). This is rather important to be clear and consistent on this through the paper because it impacts the clarity for the reader and your further interpretation/discussion.

We do not believe that the CAS concentration of our cultures grown in seawater can be set apart from the overall data set. When this variability, from one experiment to the other in the same 28.2 mM concentration (in NSW and in ASW) is taken into account, the plateau clearly starts at 28.2.

We believe that further experiments will be needed to evaluate more precisely how CAS record sulfate concentration between 28 and 40Mm.

Minor and supplementary/more detailed comments are annotated in the following pdf file.

We would like to thank Julien Richirt for these thorough revisions, we addressed all of these corrections.

**Reviewer David Evans**

This is my second review of the manuscript Impacts of seawater sulfate concentration on sulfur concentration and isotopic composition in calcite of two cultured benthic foraminifera by Thaler et al., which presents S/Ca and δ34S measurements of two strains of rosalinid foraminifera cultured under different seawater [SO4 2- ]. The authors have, in my view, adequately addressed all of the comments from the first round of review, resulting in an improved version of what was already a very interesting and comprehensive piece of work.

I have some further minor comments for the authors to consider but would suggest that they do so at their own discretion.

We would like to thank David Evans for his kind words, and for this second review, which helped us to improve the manuscript once again.

Main comment 1. Reproduction/growth in culture. If I understand what is written on lines 79 and sections 2.1.2/3.1, dead individuals (empty shells, or those not attached to the petri dishes) were removed, leaving only live foraminifera.

Is it therefore possible that foraminifera that did not die but remained dormant could be present in the final analyses? In most cases this is of course not important, as there was a large increase in the number of individuals during the experiment.

We believe that this is possible, but only in rare cases. In any case, they are outnumbered by the total number of foraminifera present in each analysis (more than 100).

A possible exception to this is the 60 mM experiment, in which the number increased from ~30 to ~100. Given that the interpretation of the [SO4 2- ]swS/Cashell plateau hinges on this experiment, I suggest adding a note to explain whether or not this datapoint unambiguously does not contain pre-experiment shell material, or if it could represent a mixture of shell material grown under normal seawater and experimental conditions (which could then be an alternate explanation for the plateau if the foraminifera from reproduction in the experiment were smaller).

We do not believe that the interpretation of the plateau holds on the 60mM data point: At 28mM the CAS concentration ranges from 10800 to 14200 ppm and all following CAS concentration (at 35, 40 and 60mM) remain in that range. Additionnally, considering the hundreds of new foraminifera formed in media 35 and 40, we do not believe that the plateau can be explained by the presence of pre-experiment foraminiferal shells (less than the initial number of foraminifera, that is as low at 6 individuals in these experiments, still clearly lower than the final number of hundreds of shells collected).

On a similar note, Table 1 gives the numbers of foraminifera in each experiment through time, but how many were discarded during the experiment? Does this provide evidence for multiple generations? Even in the 60 mM [SO4 2- ] experiment?

Considering 1 adult can give 20 to 40 juveniles, it is likely that only one generation of foraminifera was generated in the 60mM media (with enough first generation reproducing to go from 31 to 108 foraminifers) as illustrated with the stagnation in the population increase after 12 days.

In the 10mM experiment of set 2, starting with 6 foraminifers, even if all 6 had yield 40 juveniles, we would have only reach 240 specimens. So to explain a final amount of 1014 specimens we need at least 2 generations, which corroborates the fact the population is still increasing after 31 days.

Minor comments

1. Lines 26-29. An inhibitory kinetic effect would seem a more likely possibility to me, but I appreciate not every hypothesis can be listed in the abstract.

The inhibitory kinetic effect hypothesis is described in line 390 of discussion

2. Line 51. Sulphur and magnesium are not trace elements in seawater.

We changed the phrase to: "elements present in seawater get incorporated as traces in the biomineral structure"

3. Lines 58 and 451. I suggest using a different phrase to 'large volcanic events' as this possibly implies single eruptions, whereas Laakso et al. discuss large igneous provinces emplaced over thousands of years.

We replaced "large" by "important"

4. Lines 75-76. You could clarify that most studies that included material grown before culture attempt to account for this in some way, e.g. using size-mass relationships or labels.

We made the modification

5. Lines 117-118. Are the units mM or mmol/kg?

In mM

6. Lines 211-212. Were all samples run at the same concentration as the seawater standards? If not, does this approach potentially result in a reproducibility that is too low?

All samples were run on the instrument at a concentration similar to that of seawater. It has been demonstrated elsewhere that lower concentrations on the instrument do not affect the validity of the 34S value (Paris et al., 2013), but lower intensities do yield lower reproducibility, both as a result of counting statistics (and possibly Johnson noise) and a greater influence of the background subtraction.

7. Line 246. Please clarify which results you are referring to.

We modified the sentence: "take into account pH and DIC value nor foraminifera counts measured in that media after day 15. "

8. Lines 258-259. I think this explanation is unlikely (e.g. the DIC had increased by day 5 (Figure. 6) and there was no further increase.

We changed the sentence to: "DIC probably built up in the Petri dishes each week as the foraminifera respire".

9. Section 4.1 title. Bear in mind that there were large covariations in seawater [Na+ ] and [Cl- ]. Worth mentioning in the discussion?

We made the following modification line 127

The amount of NaCl in those two media was adjusted to keep the total salinity constant (35.06 g/L). $Na^+$ concentrations for ASW[0] and ASW[180] were 479 mM and 402 mM, respectively, representing a maximum 24% change while the $Cl^-$ concentrations were 612 mM and 175 mM, respectively, representing a maximum 71 % change, for a maximum 180% change in sulfate concentration.

10. Section 4.2. At some point (in the introduction?) it would be helpful to mention the Mg/Ca of these foraminifera if it is known. If they reduce the Mg/Ca of the biomineralisation site compared to seawater then the considerations regarding the effect of seawater [SO4 2- ] on CaCO3 nucleation and precipitation will likely not apply/be more complicated than implied in some places in the manuscript (e.g. lines 338, 407).

We don't know the Mg/Ca of the tests. This experiment was focused as a first step on $[SO_4^{2-}]$ and we did not measure Mg/Ca ratio, in future work it might be indeed interesting to measure the two as suggested.

11. Line 338. On a similar note, I would suggest rephrasing this sentence. There may have been no precipitation in those experiments but it does not mean it is not possible, e.g. if higher degrees of oversaturation were to be achieved.

Although we agree that increasing the saturation of seawater to favor calcite precipitation might work, it would be a completely different system.

Line 338 we state "It is remarkable to note that foraminifera can reproduce and thus calcify at $[SO_4^{2-}]$ as high as 90 mM (Fig. 5), concentrations at which no inorganic calcite precipitation occurs (Bots et al., 2011; Barkan et al., 2020). However as discussed before, their reproduction is limited to the first week, which strongly suggests that they could only tolerate brief exposure to such a high level of sulfates in their environment. "

Our sentence refers to systems with defined concentrations described in papers that are referenced, hence the "concentration space" where our sentence applies is given. We thus decided to keep the sentence as is.

Likewise, I would not read anything into the 'fact that calcite precipitates' (line 369).

There is indeed calcite precipitating in our experiments, which constitute our fact. We then use that fact as a suggestion for a mechanism "The mere fact that calcite precipitates therefore suggests that sulfate is at least partially removed from the precipitating fluid"

We believe that stating that it is only " suggesting" is showing enough caution and we decided to keep the sentence.

12. Line 355. Please clarify, the same as what? Heterostegina?

We have now clarified that the presence of sulphated glycosaminoglycans has been documented in the organic matrix test of many foraminiferal taxa (see Langer 1992) and not only in *Heterostegina* (see Weiner and Erez, 1984). In line 355 we refer rather to the bilayer test construction in all rotaliids, not just *Heterostegina*.

We specified:

The organic matrix of the test of a wide variety of foraminiferal taxa contains over-sulfated glycosaminoglycans and proteins (Weiner and Erez,1984; Langer 1992). The benthic foraminifera Rosalinidae belong to the order Rotaliida and likely share the same mechanisms of biomineralisation and bilayer test construction as other rotaliid families.

13. Lines 416-418. Given that you include speciation modelling, can you say which ion pairs become relatively more abundant?

The relative abundance stay the same, NaSO4>MgSO4>>CaSO4. Here are the endmembers of our results for two different pHs (8.2 and 8.19) at 5mM of SO42- are (in mol/L):

| CaSO4 | NaSO4⁻ | MgSO4 |
|---|---|---|
| 2E-04 | 0.002 | 0.001 |
| 2E-04 | 0.002 | 0.001 |

While at 60mM at pH 8.2 and 8.15 we get :

| CaSO4 | NaSO4⁻ | MgSO4 |
|---|---|---|
| 0.002 | 0.019 | 0.015 |
| 0.002 | 0.0189 | 0.015 |

All these results are provided in the appendix,

14. Line 720. It also diffuses between the experiment and atmosphere.

While we agree that $CO_2$ diffuses between the experiment and atmosphere, we believe that no strong enough variation in the atmospheric level of $CO_2$ (that can vary in a lab depending on how many people are working in the same room) can explain our DIC concentration changes. Additionnally, to the best of our understanding, a strong $CO_2$ variation in the atmosphere could lead to a DIC increase, correlated to a pH change, which is not what we observe.

Typos

1.Line 65. Change 'interrogates' to 'suggests'. Done

2. Lines 137. Experiments. Done

3. Line 184. Aliquots, or delete 'of the'. Done

4. Lines 314-315, 390. Change four instances of 'Mm'. Done

5. Line 693. Forms. Done

6. Line 728. On a pool of a hundred to…'. Done

7. Line 816 and Fig. D1, it should be CaHCO3 + (also NaSO4 - ). Done

---

## Author Response (AR3)

by Chiara Borrelli
**Public justification (visible to the public if the article is accepted and published)**:
The authors addressed the last suggestions provided in a satisfactory manner. The manuscript is now ready for publication in BG.

We are pleased to learn that our study can be published in BG and hope that the technical corrections we made are sufficient.

I only have two technical corrections, as specified below:

Line 145: please briefly mention the criteria used distinguish living individuals while counting, if different from the specimens being attached to the substrate.

We modified the text to explain more precisely how live individuals were counted, and the associated bias:

"Each week, live individuals were counted in each environment. As the species under study remains attached to the substrate when alive, individuals that no longer adhered to the Petri dishes, and therefore floated, were considered dead and not counted. However, it should be noted that rare dead individuals, which can sometimes be identified as unequivocally empty tests or as individuals without reticulopodial activity and/or change in color for several days (Bernhard, 2000), may remain attached and might have been counted as alive. At the same time, a few living adults may also have become detached from the substrate, and were therefore not counted as alive. As these cases are rare, the error generated by these two phenomena is largely covered by the error bar on the count in growing samples, while in samples not growing from the inoculum, living cells remain estimated."

Suppl. File: please add a caption to every figure and table included.

We made the following changes :
Fig. C1 : We added the size of the scale bar in the caption
Figure D1 : We specified that dots color indicate each sulfate concentration
Table D1 to D6 (Excell additional file): a caption was written for each table